# OMS: One More Step Noise Searching to Enhance Membership Inference Attacks for Diffusion Models

## Abstract

The data-intensive nature of Diffusion models amplifies the risks of privacy infringements and copyright disputes, particularly when training on extensive unauthorized data scraped from the Internet. Membership Inference Attacks (MIA) aim to determine whether a data sample has been utilized by the target model during training, thereby serving as a pivotal tool for privacy preservation. Current MIA employs the prediction loss to distinguish between training member samples and non-members. These methods assume that, compared to non-members, members, having been encountered by the model during training result in a smaller prediction loss. However, this assumption proves ineffective in diffusion models due to the randomly noise sampled during the training process. Rather than estimating the loss, our approach examines this random noise and reformulate the MIA as a noise search problem, assuming that members are more feasible to find the noise used in the training process. We formulate this noise search process as an optimization problem and employ the fixed-point iteration to solve it. We analyze current MIA methods through the lens of the noise search framework and reveal that they rely on the first residual as the discriminative metric to differentiate members and non-members. Inspired by this observation, we introduce **OMS**, which augments existing MIA methods by iterating **O**ne **M**ore fixed-point **S**tep to include a further residual, i.e., the second residual. We integrate our method into various MIA methods across different diffusion models. The experimental results validate the efficacy of our proposed approach.

## 1 Introduction

Recently, diffusion models (Ho et al., 2020; Song et al., 2020b) have been widely recognized for their unparalleled capability to generate images of exceptional quality, which are increasingly becoming indistinguishable from their real-world counterparts. Due to the high quality of images generated by diffusion models, an increasing number of AI companies are developing generative tools predicated on diffusion models for commercial art design.

Nonetheless, these advancements are accompanied by inherent challenges. The data-intensive nature of diffusion models has amplified the risk of privacy infringements and copyright disputes. Trained on extensive unauthorized data scraped from the Internet, these methods overlook the copyrights and privacy of the original owners. A case in point is the recent lawsuit filed by Getty Images against Stability AI, alleging unauthorized use of 12 million of Getty's images for model training (Brittain, 2023). Thus, it is imperative to develop effective tools to detect diffusion models' privacy infringements.

To audit these privacy risks, Membership Inference Attacks (MIA) (Shokri et al., 2017) have emerged as a potential solution. The objective of MIA is to ascertain whether a data sample has been utilized in the training process of a machine learning model. Existing MIA methods (Sablay-rolles et al., 2019; Salem et al., 2019; Song & Mittal, 2021) typically operate under the assumption that member records tend to exhibit lower prediction losses compared to non-member records. Consequently, these methodologies compute the prediction losses and utilize this metric to differentiate between member and non-member records.

Although the utilization of prediction loss to differentiate between member and non-member records has been empirically validated for numerous deterministic models, such as classification models and Generative Adversarial Networks (GANs) (Chen et al., 2020; Hayes et al., 2019; Hilprecht et al., 2019; Choquette-Choo et al., 2021; Hanzlik et al., 2021), its efficacy is diminished when applied to diffusion models due to the intractability of the training loss. More precisely, during the training process of the diffusion model, a random noise is sampled, serving not only as a component of the model's input but also as the target in the training loss. However, during the execution of the membership inference, it is virtually impossible to replicate the exact noise sampled during the training phase. The discrepancy between the noise used during training and membership inference contributes to the inaccuracy of the loss estimation.

To address these issues, instead of the loss assumption, we propose an alternative hypothesis: **it is more feasible for members to find the noise counterpart used in the training process**. This assumption aligns more closely with the inherent stochastic characteristics of diffusion model training. Based on this assumption, we introduce a membership inference framework tailored to diffusion models, leveraging a noise searching mechanism. We formalize the noise searching process as an optimization problem with the training loss as the optimization objective.

Moreover, we propose to utilize the fixed-point iteration to solve the optimization problem. By iteratively applying a function to the initial guess, we strive to facilitate the convergence to the noise encountered by the members during the training stage. We begin with an empirical analysis focuses on the convergence properties of the fixed-point iteration. We discern a distinct attribute where member samples exhibit faster convergence rate compared to non-member samples. This observation implies that the convergence rate can be employed as a discriminative feature to differentiate between member and non-member samples.

This attribute provides further insights into current MIAs for diffusion models. Specifically, from the perspective of the fixed-point iteration, we reinterpret current MIA methods as assessments of convergence rate, primarily through the first residual. To refine this measurement and capture the nuances of convergence dynamics more comprehensively, we introduce an augmentation to the iteration process, incorporating an additional step that considers the second residual. We term this extension the "**O**ne **M**ore **S**tep" (**OMS**) approach.

We conduct experiments across various diffusion models, spanning CNN-based and Transformer-based architectures, along with various datasets and MIA methods. Notably, to the best of our knowledge, we are the first to evaluate MIA performance on Transformer-based diffusion models. The experimental results demonstrate the effectiveness of the proposed **OMS** approach and the noise searching-based MIA framework. In summary, our paper makes the following contributions:

- We reveal the noise inconsistency issues in current MIA methods for diffusion models. To address this, we devise a novel framework in the perspective of noise searching. Formally, we conceptualize the noise searching process as an optimization problem.

- We propose the fixed-point iteration to solve the noise searching optimization problem. Moreover, we investigate its convergence properties in practice and find that members exhibit faster convergence rate compared to non-members.

- We analyze existing MIA methods through the proposed framework, revealing that the efficacy of existing methods is linked to the convergence rate, particularly as characterized by the first residual. Motivated by this, we introduce a refinement strategy by iterating **O**ne **M**ore **S**tep (**OMS**) to include the second residual.

- We conduct experiments on various diffusion models, encompassing CNN-based and Transformer-based architectures, using various datasets. The results not only confirm the validity of our MIA framework but also underscore the efficacy of the OMS refinements.

## 2 RELATED WORK

**Membership Inference Attack (MIA).** The Membership Inference Attack (MIA), initially introduced by Shokri et al. (2017), is a technique designed to extract privacy information from machine learning models. Its primary objective is to predict the presence of a specific data record in the training set of a given model. The effectiveness of MIA fundamentally relies on the hypothesis that

machine learning models exhibit differential responses to member records versus unfamiliar non-member records. Given the manner to exploit model's reactions, existing methods can be divided into two categories, model-based methods and metric-based methods. In the realm of model-based methods (Shokri et al., 2017; Salem et al., 2019; Long et al., 2020; Chen et al., 2020; Truex et al., 2019), a shadow model is trained to mimic the responses of the target machine learning model. Subsequently, attack algorithms are formulated, predicated on the reactions of the shadow model, with the ultimate objective of achieving generalization to the target model. For instance, Shokri et al. (2017) employ multiple shadow models to augment the attack success rate of the MIA task. Salem et al. (2019) discerns that the success of the shadow model is attributable to the transferability of the target machine learning model's output distribution. Long et al. (2020) initially select vulnerable (outlier) samples and find that these outlier samples are subject to a heightened privacy risk by shadow model-based attacks. Despite the significant advancements in the field, model-based methods are characterized by their computational intensity and exhibit susceptibility to alterations in the model's architecture.

Methods grounded in metrics (Sablayrolles et al., 2019; Yeom et al., 2018; Salem et al., 2020; Bentley et al., 2020) primarily employ a metric (typically the loss value) as a representative measure of the model's response to each sample. The membership of a specific sample is subsequently determined based on the numerical values of the selected metric. For example, Sablayrolles et al. (2019) delve into the investigation of the metric within a white-box context, concluding that the most effective metric is the training loss function. Yeom et al. (2018) leverage the average training loss as their chosen metric and present a discussion on the interplay between model overfitting and the performance of membership inference. Bentley et al. (2020) examine the relationship between the generalization gap and membership inference, positing that a deficiency in generalization escalates the risk of model privacy leakage.

**MIA for diffusion models.** Given the computational intensity of training a shadow model with comparable parameters, model-based methods are deemed unsuitable for diffusion models. As a result, most current MIA tailored for diffusion models (Duan et al., 2023; Matsumoto et al., 2023; Kong et al., 2023) are metric-based methods. These methods share a core assumption with MIAs applied to other models, which assumes that the loss value for members is smaller than that for non-members. A significant contribution to this field is made by Matsumoto et al. (2023), who introduce a pioneering MIA method for diffusion models. This approach utilizes the training loss of the model and identifies a timestep at which the divergence of loss between members and non-members is maximized, thereby optimizing the performance of membership inference. SecMI (Duan et al., 2023) exploits the approximated posterior error as a proxy to estimate the training loss, which subsequently serves as the membership inference metric. PIA (Kong et al., 2023) endeavors to reconstruct the training loss through the complete sampling path, thereby leveraging the training loss throughout the entire diffusion process.

Despite achieving substantial performance, these methods still suffer from the inaccuracy of loss estimation. During the training phase of the diffusion model, the loss value is dictated by the training target (a random Gaussian noise). However, when executing the MIA, replicating the exact noise sampled during the training process is virtually unattainable. Instead, we propose a novel MIA framework predicated on noise searching. This innovative approach promises to enhance the overall performance of MIA and provide insight into the principles of MIA methods for diffusion models.

## 3 METHOD

Given a data record $x_0$, the goal of Membership Inference Attacks (MIA) is to identify whether $x_0$ is in the training set of the target diffusion model $\epsilon_\theta$. Existing MIAs (Duan et al., 2023; Matsumoto et al., 2023; Kong et al., 2023) tailored for diffusion models predominantly assume that the loss values for members are lower than those for non-members. However, these loss-based approaches are susceptible to inaccuracies in loss estimation, which are caused by noise inconsistency between the training and inference stages (Section 3.2). In contrast, we propose a novel MIA framework that employs noise searching, an approach we believe aligns more closely with the stochastic nature of the model's training process. We formulate the noise searching process as an optimization problem and propose to use the fixed-point iteration to solve this problem (Section 3.3). We posit that members can retrieve the training noise with less effort compared to non-members. Subsequently,

we analyze the convergence properties of the fixed-point iteration and further validate that it is more feasible for members to search the noise than non-members (Section 3.4). Motivated by this insight, we reinterpret the underlying mechanisms contributing to the efficacy of existing MIA methods and propose an enhancement by incorporating **O**ne **M**ore iteration **S**tep (**OMS**) (Section 3.5).

### 3.1 BACKGROUND AND NOTATIONS

We begin with a brief introduction of the background and notations of the diffusion models. Denoising Diffusion Probabilistic Models (DDPM) (Ho et al., 2020; Song et al., 2020a) consist of a forward and a reverse process. The forward process, also named as the diffusion process, gradually adds Gaussian noise to the input image $x_0$ in $T$ time steps according to a predefined variance schedule $\beta_1, ..., \beta_T$:

$$q(x_t|x_{t-1}) = \mathcal{N}(x_t; \sqrt{1-\beta_t}x_{t-1}, \beta_t\mathbf{I}) \tag{1}$$

Let $\alpha_t = 1 - \beta_t$ and $\bar{\alpha}_t = \prod_{s=1}^{t} \alpha_s$, this process can be simplified to:

$$q(x_t|x_0) = \mathcal{N}(x_t; \sqrt{\bar{\alpha}_t}x_0, (1-\bar{\alpha}_t)\mathbf{I}) \tag{2}$$

When $t$ is large enough, the $\bar{\alpha}$ is approaching 0, making $x_t$ an isotropic Gaussian noise. The reverse process aims to recover the data distribution from the Gaussian noise. The reverse process in one step can be represented as:

$$p_\theta(x_{t-1}|x_t) = \mathcal{N}(x_{t-1}; \mu_\theta(x_t, t), \Sigma_t) \tag{3}$$

where $\Sigma_t$ is a constant depending on the variance schedule $\beta_t$ and $\mu_\theta(x_t, t)$ is determined by a neural network:

$$\mu_\theta(x_t, t) = \frac{1}{\sqrt{\alpha_t}}(x_t - \frac{\beta_t}{\sqrt{1-\bar{\alpha}_t}}\epsilon_\theta(x_t, t)) \tag{4}$$

By recursively leveraging the reverse step, Gaussian noise can be recovered to the original image. To train the DDPM, an image $x_0$, a timestep $t$ and a random noise $\epsilon \sim \mathcal{N}(0, \mathbf{I})$ are first sampled. A noisy image $x_t$ is then obtained by using the forward process (Equation 2). We then input both the noisy image $x_t$ and the timestep $t$ into a U-Net (Ronneberger et al., 2015) $\epsilon_\theta$ to predict the noise within $x_t$. The optimization objective for the denoising U-Net can be written as:

$$\mathcal{L} = \mathbb{E}_{t,x_0,\epsilon}[||\epsilon - \epsilon_\theta(x_0, t, \epsilon)||_2^2] \tag{5}$$

### 3.2 NOISE INCONSISTENCY BETWEEN TRAINING AND INFERENCE

The diffusion model's training procedure can be described by Equation 5. To elaborate, given the input image $x_0$ and a specific timestep $t$, a random noise $\epsilon_{train}$ is sampled from the standard normal distribution. This noise is then utilized to perturb $x_0$ into a corrupted version $x_t$, following the schedule predefined in Equation 2. Subsequently, the diffusion model, parameterized by $\theta$ generates a prediction of the noise within $x_t$ (denoted as $\epsilon_\theta(x_0, t, \epsilon_{train})$). The training loss for the diffusion model is computed as the distance between the predicted noise $\epsilon_\theta(x_0, t, \epsilon_{train})$ and the actual sampled noise $\epsilon_{train}$. During the inference phase, due to the infeasibility of the training noise $\epsilon_{train}$, an alternate noise $\epsilon_{inf}$ is sampled to estimate the loss value. However, it is important to note that there exists no guarantee that $\epsilon_{inf}$ is identical or approximately similar to the noise $\epsilon_{train}$. This inconsistency in noise significantly impacts the accuracy of the loss values, consequently affecting the effectiveness of existing Membership Inference Attacks (MIA) targeting diffusion models (Duan et al., 2023; Matsumoto et al., 2023; Kong et al., 2023).

### 3.3 MIA BY NOISE SEARCHING

In contrast to existing MIA that rely on the randomly sampled inference noise as a surrogate for the training noise $\epsilon_{train}$ to approximate the loss, thereby encountering the noise inconsistency issue, we introduce a novel MIA framework for diffusion models, leveraging a noise search strategy. Our approach aims to reconstruct, for a given record $x_0$, the corresponding training noise $\epsilon_{train}$ that minimizes the training loss as defined in Equation 5. We assume that **it is more feasible for members to obtain the training noise** $\epsilon_{train}$. We formulate the process of noise searching as an optimization problem:

$$\min_{\epsilon} ||\epsilon - \epsilon_\theta(x_0, t, \epsilon)||_p$$
$$s.t. \quad \epsilon \sim \mathcal{N}(0, \mathbf{I}) \tag{6}$$

This optimization framework directly addresses the noise inconsistency issue by focusing on identifying the true training noise $\epsilon_{train}$. Compared with loss-based methods, the proposed approach emphasizes the identification of $\epsilon_{train}$. We argue this approach aligns more coherently with the inherent stochastic nature of the diffusion model's training process.

**Fixed-point iteration.** We address the aforementioned optimization problem by the fixed-point iteration. For a given record $x_0$ and timestep $t$, the predicted noise $\epsilon_\theta(x_0, t, \epsilon)$ is solely dependent on $\epsilon$. This dependency can be represented as an implicit function $\epsilon = f(\epsilon)$. The optimal noise, which pairs with the record $x_0$ during the training process, is also identified as the solution to the implicit function $f$. To address this, we employ the fixed-point iteration (Smart, 1980). The iterative process can be represented as follows:

$$\epsilon^n = f(\epsilon^{n-1}), \quad n = 1, 2, ... \tag{7}$$

We assume that the fixed-point iteration process essentially satisfies the constraints embedded within the optimization problem, given the fact that the model $\epsilon_\theta$ is trained to generate noises adhering to the distribution. Consequently, we hypothesize that the model's outputs also conform to the standard normal distribution. Note that we do not use more advanced methods for solving implicit functions such as Newton-Raphson or Conjugate Gradient (Nocedal & Wright, 1999). This is because the Newton-Raphson method needs to compute gradients while the Conjugate Gradients need to search high dimension gradients, both of which are computationally intensive and potentially intractable.

### 3.4 CONVERGENCE OF THE FIXED-POINT ITERATION

The primary concern about the fixed-point iteration lies in its practical convergence properties. We present an empirical analysis to address this concern. Given the initial $\epsilon^0$, our objective is to demonstrate that the sequence $\{\epsilon^n\}, n \to \infty$ generated by Equation 7 converges. To achieve this, we aim to prove that the residual $\delta^n = \epsilon^n - \epsilon^{n-1}$ converges, which would imply that $\{\epsilon^n\}$ is the Cauchy sequence. The residual can be expressed as follows:

$$\begin{aligned}
||\delta^{n+1}|| = ||\epsilon^{n+1} - \epsilon^n|| &= ||f(\epsilon^n) - f(\epsilon^{n-1})|| \\
&= ||f(\epsilon^{n-1}) + \frac{\partial f(\epsilon)}{\partial \epsilon}|_{\epsilon=\epsilon^{n-1}} \cdot \delta^{n-1} + \mathcal{O}(||\delta^{n-1}||^2) - f(\epsilon^{n-1})|| \\
&\leq ||\frac{\partial f(\epsilon)}{\partial \epsilon}|_{\epsilon=\epsilon^{n-1}}|| \cdot ||\delta^n|| + \mathcal{O}(||\delta^n||^2)
\end{aligned} \tag{8}$$

In a sufficiently confined domain, the term $||\mathcal{O}(\delta^2)||$ can be considered negligible, and the convergence dynamics are primarily governed by the Jacobian norm $||\frac{\partial f(\epsilon)}{\partial \epsilon}||$. If the Jacobian norm is below 1, it indicates that the implicit function $f$ is contractive, leading to an exponential decay in the residuals, thereby affirming the convergence of the fixed-point iteration. We visualize this Jacobian norm (the top row) along with the residuals (the bottom row) across various iterations and timesteps in Figure 1. Notably, the Jacobian norm consistently remains below the threshold of 1, thereby empirically validating the convergence of the fixed-point iteration. In the residual plots, the theoretical distance (i.e., the distance between two random Gaussian noise) is depicted in green, whereas the residuals for member and non-member sets are depicted in blue and red, respectively. Moreover, it is observed that the residuals for both member and non-member sets exhibit rapid convergence, with the member set residuals smaller than those of non-member set, further indicating that it is more feasible for members to obtain the training noise. It is also important to highlight that the first residual $\delta^1$, representing the divergence between $\epsilon^0$ and $\epsilon^1$, is frequently employed as an estimation of the training loss in current MIA methods. More validation about the convergence property and convergence speed can be found in Appendix B.

### 3.5 EXISTING MIAs AND ONE MORE STEP (OMS)

We re-assess the efficacy of prevailing MIA methods through the lens of the fixed-point iteration. Current MIA methods approximate the training loss of diffusion models by measuring the divergence between the initial noise $\epsilon^0$ and the noise after the first iteration $\epsilon^1$. In the context of the fixed-point iteration, this divergence is equivalently characterized as the first residual $\delta^1$. As illustrated in the bottom row of Figure 1, this residual effectively discriminates between members and

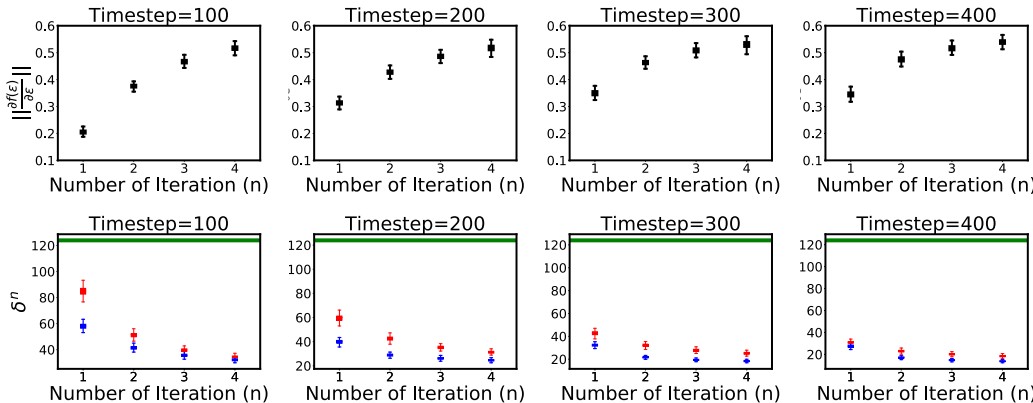

Figure 1: The top row is the Jacobian norm in different timesteps and iterations. The bottom row shows the residuals $\delta^n$, with the blue and red boxes representing the members and non-members. The green line is the theoretical distance between two random Gaussian noise. We report the results calculated over 1000 images of Cifar10 dataset (500 members and 500 non-members). The diffusion model is also trained on the Cifar10 dataset.

non-members. However, it is also discernible that the second residual $\delta^2$ for members is smaller than non-members, indicating its substantial potential for enhancing membership discrimination. Motivated by this observation, we take **O**ne **M**ore **S**tep (**OMS**) beyond $\epsilon^1$ to obtain $\epsilon^2$ and utilize the distance between $\epsilon^0$ and $\epsilon^2$ as the discriminative metric. This metric can be interpreted as an ensemble of the first and second residuals ($\delta^1$ and $\delta^2$). This approach not only preserves the discriminative capability inherent in the traditional loss-based approach (i.e., $\delta^1$), but also incorporating the extra information $\delta^2$ to augment the performance. The relationship is mathematically expressed as:

$$||\epsilon^0 - \epsilon^2|| = || \underbrace{(\epsilon^0 - \epsilon^1)}_{loss\,term} + \underbrace{(\epsilon^1 - \epsilon^2)}_{extra\,term} || = ||\delta^1 + \delta^2|| \tag{9}$$

Note that we do not leverage further residuals such as $\delta^3$ and $\delta^4$, though they also seem potential metric to distinguish the member and non-member records. This is because the marginal gain is decreased. We also provide experiments utilizing different residuals in Section 4.5. A detailed description of how the OMS is integrated with current MIA method is provided in the Appendix E for further reference.

## 4 EXPERIMENT

### 4.1 EXPERIMENTAL SETUP

**Diffusion Models and Datasets.** We evaluate our proposed method across diverse diffusion models, specifically DDPM (Ho et al., 2020), Stable Diffusion (Rombach et al., 2022) and U-ViT (Bao et al., 2023). DDPM represents a foundational approach in the realm of diffusion models, which employs convolutional neural networks as the backbone. We train DDPM on four datasets: Cifar10, Cifar100 (Krizhevsky et al., 2009), LFW (Huang et al., 2008) and Lsun-Cat (Yu et al., 2015). The Stable Diffusion models, which are prominently recognized for their text-to-image synthesis capabilities, have undergone numerous iterations. We selectively adopt SD1.5 and SD2.1 due to their widespread usage and recognition within the research community. U-ViT, a recently introduced diffusion model, incorporates transformers as its core architecture. Our investigation leverages the open-source implementation of U-ViT which has been trained on the Cifar10 datasets. We also provide more details about the diffusion models and datasets in Appendix C.

**Evaluation Metrics.** To evaluate the performance of our proposed method, we have adopted established metrics in previous works (Carlini et al., 2022; 2023; Duan et al., 2023; Kong et al., 2023) including Attack Success Rate (ASR), AUC and the True Positive Rate (TPR) at extremely low False Positive Rate (FPR). Specifically, TPR@1%FPR and TPR@0.1%FPR refer to the True Positive Rate (TPR) when the False Positive Rate (FPR) is constrained to 1% and 0.1%, respectively.

Table 1: The ASR and AUC metrics for existing MIA methods on DDPM, both with and without the integration of the One More Step (OMS). The symbol $\Delta$ is employed to denote the improvement in performance resulting from the integration of the OMS procedure.

| Method | Cifar10 | | Cifar100 | | LFW | | LSUN-Cat | | Ave | |
|---|---|---|---|---|---|---|---|---|---|---|
| | ASR | AUC | ASR | AUC | ASR | AUC | ASR | AUC | ASR | AUC |
| NA | 71.86 | 78.28 | 75.97 | 82.18 | 72.40 | 79.54 | 62.60 | 67.02 | 70.71 | 76.75 |
| +OMS | 78.21 | 85.71 | 81.49 | 88.27 | 83.02 | 90.73 | 68.89 | 75.10 | 77.90 | 84.95 |
| $\Delta \uparrow$ | **+6.35** | **+7.43** | **+5.52** | **+6.09** | **+10.62** | **+11.20** | **+6.29** | **+8.09** | **+7.19** | **+8.20** |
| SecMI | 84.01 | 90.68 | 79.83 | 86.77 | 61.49 | 64.90 | 73.13 | 79.36 | 74.62 | 80.43 |
| +OMS | 86.48 | 92.82 | 83.32 | 90.62 | 77.80 | 85.67 | 84.59 | 91.20 | 83.05 | 90.08 |
| $\Delta \uparrow$ | **+2.47** | **+2.14** | **+3.49** | **+3.85** | **+16.31** | **+20.77** | **+11.46** | **+11.84** | **+8.43** | **+9.65** |
| PIA | 88.75 | 94.89 | 85.20 | 92.21 | 82.10 | 90.17 | 77.61 | 84.87 | 83.42 | 90.54 |
| +OMS | 91.78 | 97.26 | 89.68 | 96.08 | 84.49 | 92.27 | 82.58 | 89.65 | 87.13 | 93.82 |
| $\Delta \uparrow$ | **+3.03** | **+2.37** | **+4.49** | **+3.86** | **+2.39** | **+2.11** | **+4.96** | **+4.78** | **+3.72** | **+3.28** |

Table 2: The TPR at extremely low FPR for existing MIA methods on DDPM, both with and without the integration of the One More Step (OMS). The symbol $\Delta$ is employed to denote the improvement in performance resulting from the integration of the OMS procedure.

| Method | TPR@1%FPR | | | | TPR@0.1%FPR | | | |
|---|---|---|---|---|---|---|---|---|
| | Cifar10 | Cifar100 | LFW | LSUN-Cat | Cifar10 | Cifar100 | LFW | LSUN-Cat |
| NA | 6.42 | 3.66 | 10.86 | 3.40 | 0.88 | 0.23 | 1.12 | 0.36 |
| +OMS | 12.12 | 8.03 | 30.01 | 6.37 | 2.34 | 0.66 | 5.66 | 0.79 |
| $\Delta \uparrow$ | **+5.70**$_{(+89\%)}$ | **+4.37**$_{(+119\%)}$ | **+19.15**$_{(+176\%)}$ | **+2.98**$_{(+88\%)}$ | **+1.46**$_{(+166\%)}$ | **+0.44**$_{(+191\%)}$ | **+4.54**$_{(+405\%)}$ | **+0.43**$_{(+119\%)}$ |
| SecMI | 9.15 | 7.19 | 3.65 | 3.10 | 0.49 | 0.22 | 0.42 | 0.12 |
| +OMS | 15.87 | 17.33 | 28.05 | 11.24 | 0.99 | 1.33 | 7.64 | 0.56 |
| $\Delta \uparrow$ | **+6.72**$_{(+73\%)}$ | **+10.14**$_{(+141\%)}$ | **+24.41**$_{(+668\%)}$ | **+8.14**$_{(+262\%)}$ | **+0.50**$_{(+102\%)}$ | **+1.11**$_{(+505\%)}$ | **+7.21**$_{(+1717\%)}$ | **+0.44**$_{(+367\%)}$ |
| PIA | 28.86 | 19.41 | 25.74 | 8.90 | 1.05 | 2.31 | 7.00 | 0.42 |
| +OMS | 60.11 | 48.30 | 29.72 | 13.59 | 13.24 | 10.66 | 9.22 | 0.94 |
| $\Delta \uparrow$ | **+31.25**$_{(+108\%)}$ | **+28.89**$_{(+148\%)}$ | **+3.99**$_{(+15\%)}$ | **+4.69**$_{(+52\%)}$ | **+12.19**$_{(+393\%)}$ | **+8.35**$_{(+361\%)}$ | **+2.22**$_{(+32\%)}$ | **+0.52**$_{(+124\%)}$ |

**Implementation Details.** To evaluate the effectiveness of the proposed OMS, we systematically conduct a series of experiments, aligning our benchmarks with state-of-the-art MIA methods designed for diffusion models, which include the Naive Attack (NA) (Matsumoto et al., 2023), SecMI (Duan et al., 2023), PIA (Kong et al., 2023), GSA (Pang et al., 2023) and Quantile (Bertran et al., 2024; Tang et al., 2023). We strictly follow the prescribed settings of these methods, and exclusively introduce a further fix-point iteration. Notably, our approach not only seamlessly integrates with these established MIA methods but also augments their performance. More details about these MIA methods can be found in Appendix E.

## 4.2 EVALUATION RESULTS

**Performance on DDPM.** The comparative results on DDPM, with and without OMS, are presented in Table 1. It can be observed that the OMS confers substantial improvements in performance, with increases of 8.20%, 9.65% and 3.28% in the Average AUC across the four datasets, compared to those baselines (NA, SecMI, PIA) without OMS. The improvements demonstrate the advantage of executing multiple fixed-point iterations over the conventional single-iteration approaches. We also observe that our method is particularly effective for weak attackers: an AUC increase from 64.90 to 85.67 for SecMI, from 79.54 to 90.73 for NA on the LFW dataset. Besides, we also note our method can further boost strong attackers with an average 3.72% AUC improvement for PIA. There results demonstrate the broad applicability of our proposed OMS. Furthermore, we provide the results of TPR at extremely low FPR in Table 2. These results demonstrate that the OMS notably enhances the prediction confidence, thereby amplifying the practical applicability in scenarios requiring high prediction certainty. Additionally, to provide a comprehensive visualization of the performance with and without the inclusion of OMS, we present the ROC and the log-scaled ROC curves in the Appendix D.

**Performance on text-to-image diffusion models.** Distinct from unconditional diffusion models, text-to-image diffusion models require dual inputs: the image itself and an accompanying text. How-

Table 3: Performance of existing MIA methods on text-to-image diffusion models, both with and without the integration of the One More Step (OMS). The symbol $\Delta$ is employed to denote the improvement in performance resulting from the integration of the OMS procedure.

| Method | SD1.5 | | | | SD2.1 | | | |
|---|---|---|---|---|---|---|---|---|
| | ASR | AUC | TPR@1% | TPR@0.1% | ASR | AUC | TPR@1% | TPR@0.1% |
| NA | 71.04 | 76.67 | 19.64 | 4.62 | 69.58 | 74.99 | 18.76 | 4.42 |
| +OMS | 73.34 | 79.00 | 24.40 | 8.89 | 71.45 | 77.40 | 23.14 | 8.47 |
| $\Delta \uparrow$ | **+2.30** | **+2.33** | **+4.76** | **+4.26** | **+1.86** | **+2.41** | **+4.38** | **+4.04** |
| SecMI | 57.20 | 57.60 | 6.29 | 2.04 | 57.24 | 56.61 | 3.98 | 0.78 |
| +OMS | 60.62 | 61.38 | 13.21 | 5.47 | 61.84 | 62.29 | 10.97 | 3.24 |
| $\Delta \uparrow$ | **+3.42** | **+3.77** | **+6.93** | **+3.42** | **+4.60** | **+5.69** | **+6.99** | **+2.46** |
| PIA | 63.17 | 67.59 | 12.71 | 3.76 | 71.15 | 78.38 | 18.56 | 3.36 |
| +OMS | 72.08 | 78.89 | 25.33 | 4.26 | 77.59 | 85.45 | 30.47 | 9.61 |
| $\Delta \uparrow$ | **+8.90** | **+11.30** | **+12.61** | **+0.50** | **+6.44** | **+7.07** | **+11.91** | **+6.25** |

Table 4: Performance of existing MIA methods on U-ViT, both with and without the integration of the One More Step (OMS). The symbol $\Delta$ is employed to denote the improvement in performance resulting from the integration of the OMS procedure.

| Method | Without OMS | | | With OMS | | |
|---|---|---|---|---|---|---|
| | ASR | AUC | TPR@1%FPR | ASR($\Delta$) | AUC($\Delta$) | TPR@1%FPR($\Delta$) |
| NA | 61.47 | 63.66 | 2.62 | 68.51(**+7.04**) | 74.31(**+10.65**) | 8.65(**+6.03**) |
| SecMI | 68.21 | 74.44 | 12.88 | 74.95(**+6.74**) | 82.31(**+7.87**) | 24.35(**+11.47**) |
| PIA | 54.60 | 52.91 | 1.80 | - | - | - |
| PIAN | 59.36 | 61.13 | 3.42 | 69.11(**+9.75**) | 75.42(**+14.29**) | 9.86(**+6.44**) |

ever, in real-world scenario, images are seldom annotated by texts. It is a common case that users do not have access to the text employed during the training phase. To replicate this real-world scenario, we leverage BLIP (Li et al., 2022) to generate text captions for the input images. The results on text-to-image diffusion models are detailed in Table 3. These evaluation further corroborate the substantial performance improvements that can be achieved by incorporating the OMS into current MIA methods. Additionally, we provide ROC curves and their log-scaled variants in Figure D.3.

**Performance on Transformer-based diffusion models.** The traditional diffusion models predominantly leverage CNNs as their backbone. However, recent advancements have seen an increasing trend towards the adoption of Transformers as the foundational architecture (Bao et al., 2023; Chen et al.; Peebles & Xie, 2023; Esser et al., 2024). To assess the efficacy of existing MIA methods on Transformer-based diffusion models and substantiate the effectiveness of our approach, we conduct experiments utilizing the U-ViT model, a continuous time diffusion model based on Transformers. Notably, the majority of existing MIA methods are specifically designed for discrete time diffusion models, posing a challenge for direct application to the U-ViT model. To address this, we implement a simple mapping strategy, converting the discrete timestep within the range [0, 1000] to the continuous range [0, 1]. Additionally, we observe that the performance of PIA approximates random guessing, mainly because its output distribution deviates the standard normal distribution. To mitigate this issue, we leverage a regularization technique (Kong et al., 2023), hereby referred to as PIAN. The results, presented in Table 4, reveal that the incorporation of an additional fixed-point iteration, as proposed in our method, led to performance improvements in existing methods, suggesting the robustness and efficacy of OMS approach across diffusion models with diverse architectures.

## 4.3 INTEGRATION WITH QUANTILE REGRESSION

Quantile Regression (Tang et al., 2023) incorporates the t-error metric (proposed by SecMI (Duan et al., 2023)) to learn a quantile regression model that predicts the $\alpha$-quantile of the t-error for each individual sample. This approach enables the estimation of a sample-specific $\alpha$-quantile as a re-

Table 5: Performance of OMS in Quantile Regression (QR).

| Method | DDPM-Cifar10 | | | U-ViT-Cifar10 | | |
|---|---|---|---|---|---|---|
| | TPR@5% | TPR@1% | TPR@0.1% | TPR@5% | TPR@1% | TPR@0.1% |
| QR (t-error) | 27.76 | 6.10 | 0.38 | 17.15 | 3.13 | 0.54 |
| QR (t-error+OMS) | 44.66 | 17.14 | 1.38 | 31.97 | 8.62 | 1.20 |
| $\Delta \uparrow$ | **+16.90** | **+11.04** | **+1.00** | **+14.82** | **+5.49** | **+0.66** |

Table 6: Performance of OMS in Gradient-Based Method (GSA).

| Method | SD1.5 | | | | SD2.1 | | | |
|---|---|---|---|---|---|---|---|---|
| | ASR | AUC | TPR@1% | TPR@0.1% | ASR | AUC | TPR@1% | TPR@0.1% |
| GSA | 87.56 | 94.19 | 55.33 | 21.85 | 87.94 | 94.50 | 55.86 | 21.37 |
| GSA+OMS | 88.12 | 94.53 | 56.17 | 22.18 | 88.64 | 95.11 | 58.82 | 22.53 |
| $\Delta \uparrow$ | **+0.56** | **+0.34** | **+0.84** | **+0.33** | **+0.70** | **+0.61** | **+2.96** | **+1.16** |

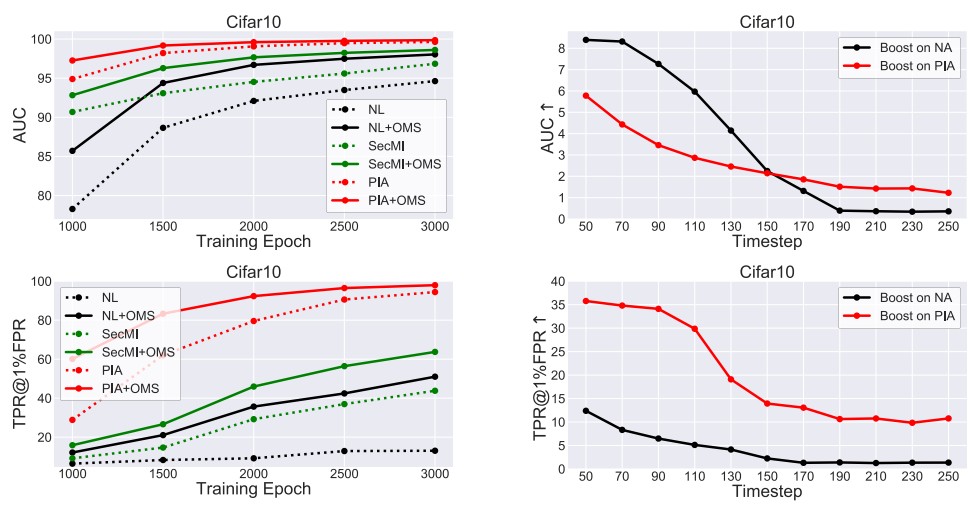

(a) The results of OMS for different training epochs.    (b) The results of OMS for different timesteps.

Figure 2: The results of AUC and TPR@1%FPR metrics of OMS for different training epochs and different timesteps.

fined per-sample threshold for identifying membership status. While t-error serves as a fundamental confidence metric for quantile regression, we have shown that the t-error can be augmented through OMS (Table 1- 3). Similarly, we refine quantile regression by incorporating an additional fixed-point iteration to current confidence metric (t-error). This refinement leads to improved performance, as evidenced in Table 5.

### 4.4 INTEGRATION WITH GRADIENT-BASED METHOD

GSA (Pang et al., 2023) constitutes a gradient-based MIA method which posits that the gradients inherently convey a more direct indication of how the target model responds to member and non-member samples. As a white-box attacker, GSA demonstrates significant efficacy against diffusion models compared to other attackers. We concentrate on the back-propagation GSA, which harnesses the backward pass of gradient computation during loss optimization to execute MIA. We refine this back-propagation GSA by backwarding the loss after OMS (Equation 7). Notably, as evidenced in Table 6, the OMS is also capable of enhancing the efficacy of gradient-based methods.

### 4.5 ABLATION STUDY

**The Training Steps.** Previous researches (Yeom et al., 2018; Leino & Fredrikson, 2020; Salem et al., 2019) have highlighted the tendency of machine learning models to memorize training data

as the training procedure progresses. In response to these insights, we have conducted an evaluation of our method throughout the training process. The results are presented in Figure 2(a). We observe that all these examined MIA methods exhibit enhanced performance as the training epochs increase, which corroborates the phenomenon of model memorization. Another notable observation is the consistent efficiency of our method throughout the entire training process. Furthermore, we identify that PIA begins to saturate in terms of AUC after 1500 training epochs. Our method further boosts PIA's performance on TPR@1%FPR, providing compelling evidence for the superiority and robustness of our approach.

**The Timesteps.** The timestep serves as a crucial parameter for the level of noise incorporated into the input of the denoising U-Net, significantly impacting the performance of the diffusion models. Consequently, we execute MIA across a range of timesteps, specifically from 50 to 250. The performance enhancements attributed to OMS are presented in Figure 2(b). Our observations indicate that the OMS exhibits robust performance across timesteps but the improvements diminishes as the timestep increases. This observed decline can be attributed to the increasing prominence of noise in the model's input. Specifically, as the timestep increases, the noise component becomes the dominant factor, potentially disrupting the stability of the fixed-point iteration process.

**The number of iteration.** We incorporate an additional fixed-point iteration for computational efficiency, which is also validated by previous experimental results. In this experiment, we explore varying fixed-point iterations and harness the distance to execute MIA. The results are depicted in Figure 3. Specifically, $||\epsilon^1 - \epsilon^0||$ represents the NA approach, whereas $||\epsilon^2 - \epsilon^0||$ represents NA with OMS in previous experiments. It is evident that increasing the number of iterations leads to improved performance, with $||\epsilon^3 - \epsilon^0||$ demonstrating the optimal results. While residuals ($||\epsilon^2 - \epsilon^1||$ and $||\epsilon^3 - \epsilon^2||$) exhibit some level of effectiveness, their

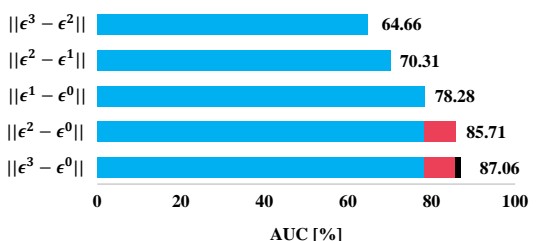

Figure 3: The results (AUC) utilizing different number of fixed-point iteration in Cifar10 dataset.

performance diminishes as the iteration count increases. It is also noteworthy that while additional fixed-point iterations hold the potential for superior performance, the marginal gains diminish progressively. This phenomenon can be attributed to the decreasing efficacy of residuals, as $||\epsilon^n - \epsilon^0||$ can be viewed as an ensemble of these residuals. For instance, $\epsilon^2 - \epsilon^0$ can be interpreted as an aggregate of $\epsilon^2 - \epsilon^1$ and $\epsilon^1 - \epsilon^0$.

## 5 CONCLUSION

In this paper, we explore the MIA for diffusion models in a novel perspective, i.e., the noise searching. We first analyze the noise inconsistency issue between the training and membership inference stage. To address this issue, we introduce a noise searching framework that formulates the search for optimal training noise as an optimization problem. Utilizing the fixed-point iteration, we solve the optimization problem and conduct a thorough examination of its convergence properties, revealing distinct convergence rates between member and non-member data. Inspired by this insight, we rethink the effectiveness of current MIA methods and propose an enhancement through one more iteration step, resulting in a substantial performance boost for existing MIA methods. The noise searching framework provides a unique and unified perspective for comprehending the fundamental principles of MIA tasks for diffusion models. We anticipate that our contributions will foster further research into the privacy risks associated with diffusion models and contribute to the ongoing research in this field.

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

## A PRELIMINARIES ON FIXED-POINT ITERATION

Fixed-point iteration is a fundamental technique employed to find the roots of equations, particularly those that are difficult to solve analytically. This method involves transforming the original equation into an equivalent fixed-point problem, $x = g(x)$. The process begins with an initial guess $x_0$ and generates a sequence of approximations $\{x_n\}$ using the iterative formula $x_{n+1} = g(x_n)$. The

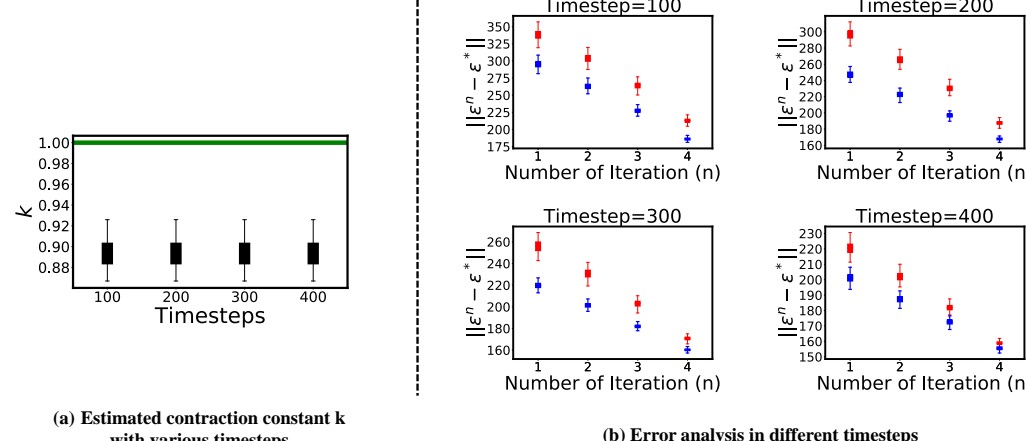

(a) Estimated contraction constant k
with various timesteps

(b) Error analysis in different timesteps

Figure B.1: (a) We estimate the contraction constant $k$ in different timesteps. The $k$ is continuously below 1, which further demonstrates the convergence of the fixed-point iteration. (b) We analyze the convergence speed from the lens of error $||\epsilon^n - \epsilon^*||$ (members in blue and nonmembers in red). The error upper bound of members is significantly lower than nonmembers, demonstrating that members converge faster than nonmembers.

convergence of this sequence is the fixed point $x^*$ where $x^* = g(x^*)$. In this paper, we formulate the noise search problem as $\epsilon = \epsilon_\theta(x_0, t, \epsilon)$ where $x_0, t$ are predefined. This formulation is further simplified to a canonical fixed-point problem, represented as $\epsilon = f(\epsilon)$.

**The Convergence Property.** The theoretical foundation for fixed-point iteration is grounded in the contraction mapping theorem (Berinde & Takens, 2007; Berinde, 2004b). This theorem states that if $g$ is a contraction mapping on a complete metric space, then $g$ possesses a unique fixed point, and the sequence generated by the iteration will converge to this fixed point for any initial guess. Specifically, a contraction mapping is defined as a function for which there exists a constant $0 \le k < 1$ such that for any $x, y$ in the space, $d(g(x), g(y)) \le k \cdot d(x, y)$, where $d$ is a metric.

## B  MORE ANALYSIS ON CONVERGENCE PROPERTY

### B.1  EMPIRICAL ANALYSIS ON CONTRACTION CONSTANT

According to the contraction mapping theorem, the convergence of fixed-point iteration is guaranteed when the objective function $f$ satisfies the contraction property. While prior research (Davydov et al., 2024; Kozachkov et al., 2022; Fazlyab et al., 2019) has investigated the contraction of neural networks, these efforts predominantly have primarily concentrated on simple networks with limited layers. To the best of our knowledge, no existing work has provided effective methods for evaluating the contraction of diffusion models which possess large-scale parameters, complex architectures, and highly non-linear properties.

### B.2  THE CONVERGENCE SPEED BY ERROR ANALYSIS

In the main text, we utilize the residual $\delta$ as a metric for assessing the convergence speed, as it represents a straightforward index to evaluate the convergence of the sequence $\{\epsilon^n\}$. In this section, we introduce another measurement of convergence speed based on the work of Berinde & Takens (2007) from the lens of error analysis. Suppose that for two fixed-point iterations $\{x_n\}$ and $\{y_n\}$, the following error estimates are available:

$$||x_n - x^*| \le a_n, \quad n = 0, 1, 2... \tag{B.1}$$

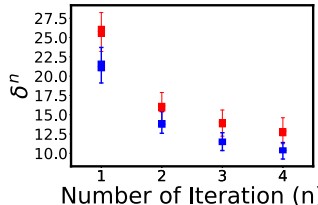 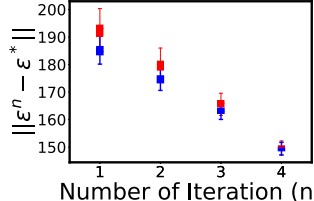

Figure B.2: The convergence speed of U-ViT measured by residual (left) and error (right). The member samples are in blue while nonmember samples in red.

and

$$||y_n - y^*| \le b_n, \quad n = 0, 1, 2... \tag{B.2}$$

where $\{a_n\}$ and $\{b_n\}$ are two sequences of positive real numbers converging to zero. Then, we state the following theorem:

**Theorem B.1** *If $\{a_n\}$ converges faster than $\{b_n\}$, then the fixed-point iteration $\{x_n\}$ converges faster to $x^*$ than the fixed point iteration $\{y_n\}$.*

To estimate the error $||\epsilon^n - \epsilon^*||$ for both members and nonmembers, we compute the error over 1000 images (500 members and 500 nonmembers) from the Cifar10 dataset. Since the optimal $\epsilon^*$ (where $\epsilon^* = f(\epsilon^*)$) is not accessible, we use $\epsilon^{20}$ where $\epsilon^{20} \approx f(\epsilon^{20})$ as a proxy for $\epsilon^*$. The results are shown in Figure B.1(b). The blue boxes represents errors for members while the red box for nonmembers. The results indicate that member errors are significantly smaller than nonmember errors, exhibiting a lower upper bound. This experimental evidence further supports the claim that members exhibit a faster convergence speed than nonmembers. We note that the error $||\epsilon^n - \epsilon^*||$ seems a potential metric to distinguish members and nonmembers. However, leveraging the error as a metric is not efficient due to the need for numerous iterations to estimate $\epsilon^*$. Therefore, we utilize the residual, as in the main text, to measure convergence speed.

### B.3 CONVERGENCE ANALYSIS ON ViT-BASED DIFFUSION MODELS

We supplement our convergence analysis with results on ViT-based diffusion models. We compute the error and the residual and present the results in Figure B.2. The blue boxes represent for members while the red boxes for nonmembers. The experimental results demonstrate that the assumption that members converge faster than nonmembers also holds for ViT-based diffusion models, which reinforces the theoretical foundation of our approach.

## C MORE DETAILS ON EXPERIMENTAL SETUPS

Table C.1: The diffusion models, resolutions, member sets and non-member sets utilized in the evaluation process.

| Diffusion Model | Resolution | Member Set | Number | Nonmember Set | Number |
|---|---|---|---|---|---|
| DDPM-Cifar10 | 32 | Cifar10 | 25,000 | Cifar10 | 25,000 |
| DDPM-Cifar100 | 32 | Cifar100 | 25,000 | Cifar100 | 25,000 |
| DDPM-LFW | 128 | LFW | 7,072 | LFW | 6,161 |
| DDPM-LSUN-Cat | 128 | LSUN-Cat | 20,000 | LSUN-Cat | 20,000 |
| SD1.5 | 256 | LAION | 5,000 | CoCo2017-Val | 5,000 |
| SD2.1 | 512 | LAION | 5,000 | CoCo2017-Val | 5,000 |
| U-ViT-Cifar10 | 32 | Cifar10 | 10,000 | Cifar10 | 10,000 |

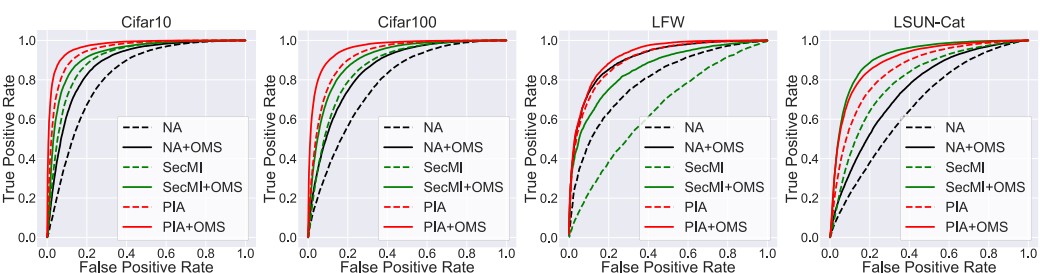

Figure D.1: The ROC curves of DDPM with and without the OMS. We employ the same color with solid/dotted line to represent methods with/without the proposed OMS.

**Diffusion Models and Datasets.** We utilize three diverse diffusion model in the evaluation process: DDPM (Ho et al., 2020), Stable Diffusion (Rombach et al., 2022) and U-ViT (Bao et al., 2023). We train DDPM on four datasets: Cifar10, Cifar100 (Krizhevsky et al., 2009), LFW (Huang et al., 2008) and LSUN-Cat (Yu et al., 2015). Both Cifar10 and Cifar100 consist of 60,000 images, segmented into a training subset of 50,000 images and a testing subset of 10,000 images. We randomly select half of the training subset (i.e. 25000 images), meanwhile ensuring an equal distribution of images across each class, to train our DDPM. For the LFW dataset, which contains approximately 13,000 images of 5,749 identities, we utilize half of the identities for training (7072 images), with the remaining identities designated as non-members. For the LSUN-Cat dataset, containing approximately 446,000 images, we select 20,000 images for training and randomly sample an additional 20,000 images from the remaining images to maintain a balanced ratio between member and non-member sets. The architecture of the DDPM follows the structure described in (Ho et al., 2020). The training epoch is set to 500 for Cifar10 and Cifar100 datasets, whereas for LFW and LSUN-Cat datasets, we train for 200 epochs. In the case of Stable Diffusion, we employ pretrained versions (SD1.5 [1] and SD2.1 [2]) trained on the LAION dataset (Schuhmann et al., 2022). We randomly sample 5,000 images for the member set and select the COCO2017-Val (Lin et al., 2014) dataset for the non-member set. For the U-ViT, we leverage open-source implementations [3]. For the Cifar10 dataset, the entire training set is appointed as the member set, with the test set serving as the non-member set.

## D    MORE EXPERIMENTAL RESULTS

To provide a comprehensive visualization of the performance with and without the inclusion of OMS, we present the ROC and the log-scaled ROC curves. Specifically, we utilize curves with dotted lines to represent MIA method without the OMS. The corresponding one with the OMS are represented with solid lines. We provide ROC curves and log-scaled ROC curves of the DDPM in Figure D.1 and Figure D.2. The ROC curves and log-scaled ROC curves of the Stable Diffusion is depicted in Figure D.3.

## E    INTEGRATION OMS WITH CURRENT MIA METHODS

### E.1    INTEGRATION WITH LOSS-BASED METHODS

We provide a comprehensive analysis of loss-based MIA methods in the context of diffusion models (Matsumoto et al., 2023; Duan et al., 2023; Kong et al., 2023). We first present the threat model of these methods. Subsequently, we present a detailed description of each method, including how these methods estimate the training loss (Equation 5) of the diffusion models. To integrate OMS with loss-based approaches, we interpret the training loss as the initial residual $\delta^1$ from the lens of

---

[1] https://huggingface.co/stable-diffusion-v1-5/stable-diffusion-v1-5

[2] https://huggingface.co/stabilityai/stable-diffusion-2-1-base

[3] https://github.com/baofff/U-ViT

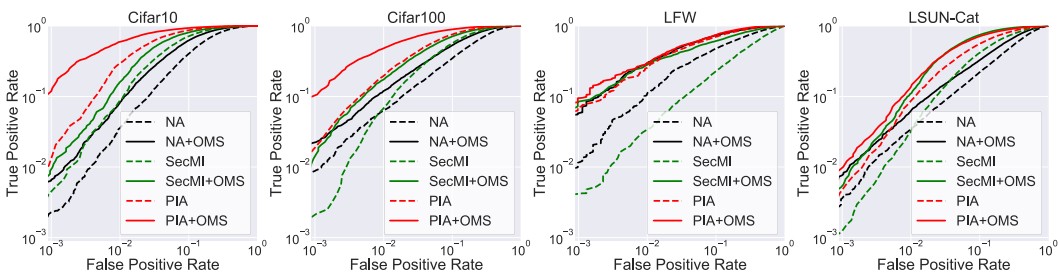

Figure D.2: The log-scaled ROC curves of ddpm with and without the OMS. We employ the same color with solid/dotted line to represent methods with/without the proposed OMS.

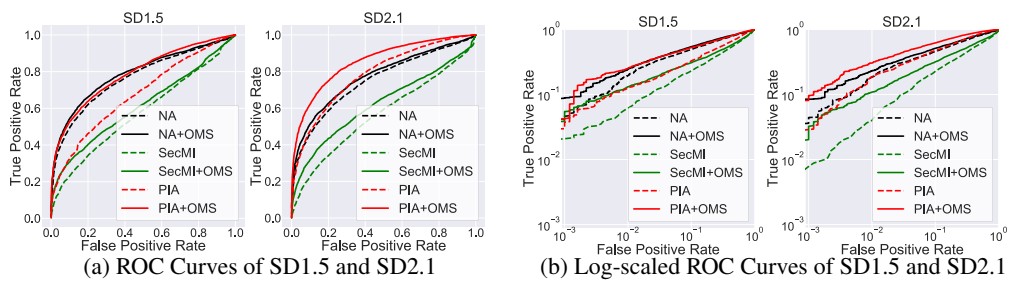

(a) ROC Curves of SD1.5 and SD2.1      (b) Log-scaled ROC Curves of SD1.5 and SD2.1

Figure D.3: The ROC (a) and log-scaled ROC (b) curves of text-to-image diffusion models with and without the OMS. We employ the same color with solid/dotted line to represent methods with/without the proposed OMS.

the noise searching framework. We introduce an additional iteration step to capitalize on the second residual (Equation 9).

**Threat Model.** Consider a target diffusion model $\epsilon_\theta$ trained on a dataset $D_{mem}$ comprising member images. The set of images not utilized during training constitutes the non-member set $D_{non}$. Given a data record $x$ and the target diffusion model $\epsilon_\theta$, the objective is to devise a method $\mathcal{M}$ to determine whether $x$ belongs to the training set $D_{mem}$. Loss-based methods assume that the loss for members is lower than that for non-members. For a specific record $x$, if the loss term $l$ falls below a predefined threshold $\tau$, $x$ is categorized as a member of the training set, labeled as '1'. Conversely, $x$ is labeled as a non-member if the loss exceeds $\tau$. This process can be expressed as:

$$\mathcal{M}(x, \epsilon_\theta) = \mathbb{1}[l(\epsilon_\theta, x_0) \leq \tau] \tag{E.1}$$

Since loss-based methods require the estimation of the training loss, which corresponds to the intermediate outpu of the U-Net, they are classified as gray-box attackers. In the gray-box setting, attacker has the access to the intermediate outputs while the model's weight remain restricted.

**Naive Attack.** Naive Attack (Matsumoto et al., 2023) represents the most straightforward approach among attackers targeting diffusion models. This attack assumes that membes exhibit lower loss values compared to non-members. The loss is estimated by utilizing a randomly sampled noise drawn from a standard normal distribution $\epsilon \sim \mathcal{N}(0, \mathbf{I})$:

$$l = |\epsilon - \epsilon_\theta(x_0, t, \epsilon)| \quad \epsilon \sim \mathcal{N}(0, \mathbf{I}) \tag{E.2}$$

where the timestep $t$ is set to 350.

**PIA.** PIA (Kong et al., 2023) leverages the divergence between sampling trajectories. This divergence is ultimately reduced to a metric based on the loss term. In DDIM (Song et al., 2020a) framework, given two points $x_0$ and $x_k$, intermediate points $x_t$ can be derived by:

$$x_t = \sqrt{\bar{\alpha}_t} x_0 + \sqrt{1 - \bar{\alpha}_t} \cdot \frac{x_k - \sqrt{\bar{\alpha}_k} x_0}{\sqrt{1 - \bar{\alpha}_k}} \tag{E.3}$$

The authors compute the initial noise when $t = 0$:

$$\bar{\epsilon}_0 \approx \epsilon_\theta(\sqrt{\bar{\alpha}_0} x_0 + \sqrt{1 - \bar{\alpha}_0} \bar{\epsilon}_0, 0) = \epsilon_\theta(x_0, 0) \tag{E.4}$$

Then, they utilize the distance of the initial point as a proxy for trajectory distance and further simplify this to:

$$l = ||\bar{\epsilon}_0 - \epsilon_\theta(x_0, t, \bar{\epsilon}_0)||_p \tag{E.5}$$

where $p = 4$ and $t = 0$. Compared to NA, PIA employs the l4 norm and uses the initial noise estimate $\bar{\epsilon}_0$ to replace the randomly sampled noise $\epsilon \sim \mathcal{N}(0, \mathbf{I})$.

**SecMI.** SecMI (Duan et al., 2023) leverages the t-error, which is defined as the distance between the authentic DDIM sampling outcomes $x_t$ and their reconstructed approximations $\tilde{x}_t$. This distance between $x_t$ and $\tilde{x}_t$ can be reduced as a scaled loss term. Initially, the $x_t$ undergoes a diffusion process to the subsequent timestep $k$:

$$\frac{x_t - \sqrt{1 - \bar{\alpha}_t}\epsilon}{\sqrt{\alpha_t}} = \frac{x_k - \sqrt{1 - \bar{\alpha}_k}\epsilon}{\sqrt{\alpha_k}}$$

$$x_k = \frac{\sqrt{\bar{\alpha}_k}}{\sqrt{\bar{\alpha}_t}}x_t + \frac{\sqrt{\bar{\alpha}_t\bar{\beta}_k} - \sqrt{\bar{\alpha}_k\bar{\beta}_t}}{\sqrt{\bar{\alpha}_t}}\epsilon \tag{E.6}$$

Subsequently, $x_k$ is denoised back to the original timestep $t$, resulting in the reconstructed $\tilde{x}_t$. The denoising process from $k$ to $t$ mirrors Equation E.6, yet the noise is estimated by the neural network $\epsilon_\theta(x_0, t_b, \epsilon)$:

$$\tilde{x}_t = \frac{\sqrt{\bar{\alpha}_t}}{\sqrt{\bar{\alpha}_k}}x_k + \frac{\sqrt{\bar{\alpha}_k\bar{\beta}_t} - \sqrt{\bar{\alpha}_t\bar{\beta}_k}}{\sqrt{\bar{\alpha}_k}}\epsilon_\theta(x_0, k, \epsilon) \tag{E.7}$$

By substituting Equation E.6 into Equation E.7, the divergence between $x_t$ and $\tilde{x}_t$ can be simplified as a scaled loss term:

$$l = ||x_t - \tilde{x}_t||_p = \frac{\sqrt{\bar{\alpha}_t\bar{\beta}_k} - \sqrt{\bar{\alpha}_k\bar{\beta}_t}}{\sqrt{\bar{\alpha}_k}}||\epsilon - \epsilon_\theta(x_0, k, \epsilon)||_p \tag{E.8}$$

where $p = 2$, $t = 100$, $k = 110$.

**The threshold $\tau$.** The threshold $\tau$ exhibits a close correlation with the attack success rate (ASR). We randomly select a small fraction (10%) of samples to obtain an optimal threshold, which is then used to classify the remaining samples. This process is repeated 10 times, and the average ASR is reported.

## E.2 INTEGRATION WITH QUANTILE REGRESSION

Quantile Regression (Bertran et al., 2024; Tang et al., 2023) computes distinct thresholds for individual samples by constructing a dataset $\{(x_i, l_i)\}_{i=1}^n$, where $x_i$ represents samples excluded from the training process and $l_i$ denotes the corresponding t-error proposed by SecMI (Duan et al., 2023). Utilizing this dataset, a quantile regression model $q$ is trained to predict $q(x)$, which is the $1 - \alpha$ quantile of the conditional distribution on t-error. Intuitively, for a given record $x_0$, if the t-error is less than the predicted $1 - \alpha$ quantile $q(x)$, it indicates that the $x_0$ belongs to the training set with a confidence level exceeding $1 - \alpha$. Adhering to the method outlined in the original paper, we leverage a CNN-based neural network as the backbone of the regression model and establish three distinct prediction heads, each corresponding to an $\alpha$ value of 0.05, 0.01, 0.001, respectively. To integrate the OMS with Quantile Regression, we leverage the t-error after one more fixed-point iteration as the $l_i$ in the dataset.

## E.3 INTEGRATION WITH GRADIENT-BASED METHOD

GSA (Pang et al., 2023) is a gradient-based MIA method which hypothesizes that the gradients inherently provide a more direct indication of how the target model responds to member and non-member samples. As a white-box attacker, GSA demonstrates significant efficacy against diffusion models compared to gray-box attackers. GSA accumulates gradients through the backpropagation of the training loss (Equation 5), across timesteps ranging from 0 to 1000, with an interval of 100. Subsequently, the gradients, collected from every component within the U-net, are subjected to their l2 norm and then flattened into a tensor of 6860 dimensions. This tensor serves as input to a pretrained machine learning model (a XGBoost (Chen & Guestrin, 2016) classifier) to predict its membership status. The XGBoost model is trained on a subset of the gradients derived from both

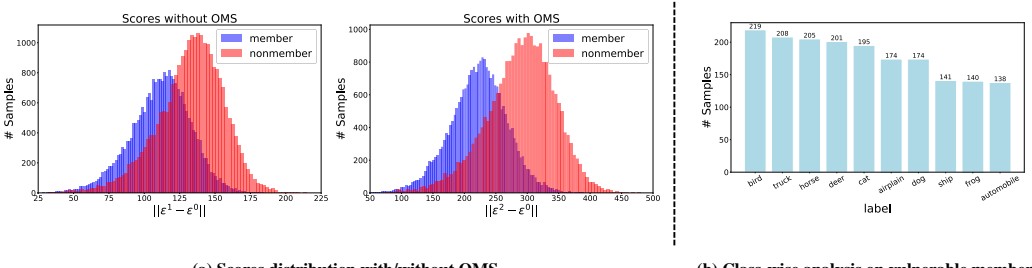

(a) Scores distribution with/without OMS

(b) Class-wise analysis on vulnerable members

Figure F.1: (a) The score distribution without/with OMS of Cifar10 dataset. (b) The label distribution of misclassified members.

member and non-member samples. To integrate the OMS with GSA, we extend the backpropagation process of the Equation 9.

## F    FURTHER ANALYSIS ON VULNERABLE MEMBERS

In this section, we endeavor to examine the effects of the OMS takes on different samples. We depict the score distribution before and after applying OMS of Cifar10 dataset in Figure F.1(a). The experimental setups are the same as those in Table 1, where there are 25,000 members and 25,000 nonmembers. The results indicate that our method significantly enhances the separability between members and nonmembers. Among all the 50,000 tested samples (25,000 members and 25,000 nonmembers), 1795 (3.59%) samples incorrectly classified as nonmembers are correctly classified as members while 1510 (3.02%) samples incorrectly classified as members are correctly classified as nonmembers, resulting in an overall 6.61% improvement. Besides, we present those 1795 misclassified members and depict their labels in Figure F.1(b). We observe that these members are nearly uniformly distributed across various class, suggesting that no significant subset of members benefits disproportionately from the additional step.

## G    LIMITATION AND FUTURE WORK

Our proposed framework suffers from two primary limitations. First, taking a high-dimensional nonlinear complex neural networks (i.e., the diffusion model) as the implicit function, there exists an absence of rigorous mathematical proofs to ensure the convergence of the fixed-point iteration process. Though we validate its convergence in practical use, this gap in theoretical understanding poses a challenge for the stability of our framework. Second, while OMS shows promise in enhancing existing MIA methods, the introduction of an additional iteration step necessitates an extra query to the target diffusion model and an increase in computational resources. This trade-off between performance enhancement and computational cost underscores the need for future research to explore more efficient iteration strategies, such as Krasnoselkij, Mann and Ishikawa iterations (Babu & Vara Prasad, 2006; Berinde, 2004a;b). We leave it for our future work.

## H    IMPLICATIONS OF MIA FOR DIFFUSION MODEL

This section presents a discussion on the practical applications of MIAs in the context of diffusion models. The implications of MIA vary depending on the stakeholders involved, namely individuals and organizations. **For individuals**, MIA poses a threat to data privacy as they allow attackers to deduce whether a particular data sample has been used to train a specific machine learning model. This can lead to the leakage or exposure of training data privacy, which is particularly alarming when the data used in model training is sensitive (e.g., medical data). Consequently, understanding MIA raises awareness about the protection of personal data privacy. Conversely, MIA can also serve as a

tool for individuals to ascertain whether their data (such as their portrait or artwork) has been used without authorization. Given that diffusion models are utilized in various applications, the detection of data abuse has become an urgent necessity. By determining whether an individual's identity is included in the training set of the target diffusion model, one can infer unauthorized usage and thereby protecting individual rights. **For organizations**, MIA can be leveraged to assess potential risks and serve as a privacy measurement. Additionally, MIA can be employed for auditing and regulatory purposes. For instance, designing an auditing model to detect the inclusion of a user's personal image in the training set of the target model enables auditing and regulation of the target system model.

## I ETHICAL STATEMENT

The primary objective of our research is to devise a method capable of discerning whether a particular sample was included in the training dataset. The proposed method offers a multitude of beneficial applications, encompassing the detection of privacy violations and the assessment of model privacy. While acknowledging the potential for malevolent entities to misuse our method for privacy attacks, we underscore the capacity of privacy protection techniques, such as differential privacy, to counteract such threats. It is crucial to note that the development of these techniques is not intended to facilitate malicious activities, but rather to advance the field of privacy protection. We trust that our contributions will be used responsibly to enhance privacy protection measures and promote ethical practices in machine learning research.

