# OpenReview forum: "OMS: One More Step Noise Searching to Enhance Membership Inference Attacks for Diffusion Models"
_ICLR.cc/2025/Conference — Submitted to ICLR 2025_

### Official Review · Reviewer_avyx · 2024-11-01

**Soundness:** 3
**Presentation:** 2
**Contribution:** 2
**Rating:** 5
**Confidence:** 3

**Summary:**

The paper proposes a way how to improve membership inference attacks (MIAs) against diffusion models (DMs). The key contribution lies in identifying that considering one step in the denoising process as a membership signal yields suboptimal results and that considering a second additional step provides a higher signal.

**Strengths:**

The experimental evaluation considers multiple model architectures and existing MIAs against DMs and shows a performance improvement over all of them. This highlights as well that the approach is fairly simple and can be integrated universally.

**Weaknesses:**

Overall, the contribution of just including an additional step into the MIAs seems comparably small. While a lot of effort it done to put the contribution into a theoretically sound presentation, it still relies on the observation made by previous MIAs that there is a noise difference between members and non-members.
I see possibilities to extend the contribution by considering the following angles:
- Analysis of disparate effect over different members: Are certain members more vulnerable? Can that be detected better with the additional step?
- Deriving formal upper bounds on the membership risk exposed by the different points.
- Analyzing why certain datasets/attack combinations benefit more from the approach.
By incorporating such an angle, the contribution could be significantly broadened.

**Experimental Results**

I would suggest, especially for table 2, not to report the delta in absolute percentages. This makes a comparison extremely unintuitive. Instead, I suggest reporting an improvement in percent. E.g., +5.70 sounds way more impressive than +0.44 in the first row, however, the first one is not even doubling while the second one is tripling the success.



**General Presentation Issues**

I would suggest fixing the following presentation issues:
- The paper does not use \citep correctly: everywhere in the intro (and nearly everywhere else in the paper), it should be \citep instead of \cite. There is also \citet. This would be used to avoid phrases as "A significant contribution to this field is made by Matsumoto et al. Matsumoto et al. (2023).
- DDPM is not introduced as an abbreviation.
- The paper introduces the abbreviation DMs but then inconsistently still uses "diffusion models", same for MIA.
- What does it mean: "During the inference phase, due to the infeasibility of the training noise" --> what is infeasible? Getting access to the original noise?
- It would be good to give an intuition on what is "fixed-point iteration" already in the intro when first mentioning it to facilitate the reader's understanding.

**Questions:**

/

---

> ### Author Response · Authors · 2024-11-19
> **Response to Reviewer avyx (1/2)**
>
> Thank you for taking the time to review our paper, and for your helpful comments. Below we will address your concerns point by point:
>
> >Weakness: Overall, the contribution of just including an additional step into the MIAs seems comparably small. While a lot of effort it done to put the contribution into a theoretically sound presentation, it still relies on the observation made by previous MIAs that there is a noise difference between members and non-members.
>
> We would like to very respectfully, but firmly, push back against the assertion that our method is only a minor variation on existing MIA. While we have no doubt that the reviewer understands these “simple” methods from previous works and our study, we believe there may be an oversight from the reviewer regarding the overall progress and current state of this research direction.
>
> Loss-based methods continue to dominate the field of MIA due to their direct measure of the model’s familiarity with training data. Consequently, most initial work on MIAs against diffusion models has also employed loss, similar to MIAs on other deterministic models. **However, these approaches fail to account for the unique randomness of diffusion models, which has not been analyzed in previous studies.**
>
> **Our method is highly motivated by the stochastic nature of diffusion models.** We reveal that the **noise inconsistency results in unreliable loss estimation** during the inference stage. We further ask **why these unreliable losses can also achieve certain MIA performance.** We answer this question by our proposed noise search framework. We reveal that **the convergence speed, rather than the loss itself, is the key factor distinguishing members and nonmembers: members and nonmembers exhibit completely different convergence dynamics in the noise search process.** We further analyze previous loss-based MIAs from the perspective of our proposed noise search framework and show that **the “loss” (i.e., the noise difference) is actually a residual, indicating the convergence speed.** Based on this analysis, we propose OMS.
>
> We believe our method is distinct from existing MIA and our contribution is significant. Besides, **we provide a deeper exploration of the convergence property in Appendix B.** Specifically, we validate the convergence of the fixed-point process **using the contraction mapping theorem**, analyze the convergence speed through **error analysis**, and expand the validation of these convergence property to transformer-based diffusion models. In summary, we believe our contribution is significant and represents a meaningful advancement in the field of MIAs against diffusion models.
>
> >Weakness: Analysis of disparate effect over different members: Are certain members more vulnerable? Can that be detected better with the additional step?
>
> Thanks for this advice! We supplement an analysis on the effects of the OMS on different samples. More specifically, we depict the score distribution before and after applying OMS to visualize the impact brought by OMS (Figure F.1(a)). Furthermore, we examine the label distribution of the 1795 members that are misclassified as nonmembers before OMS (Figure F.1(b)). **The results indicate that these members are nearly uniformly distributed across various classes, suggesting that no significant subset of members benefits disproportionately from the additional step.**

---

> ### Author Response · Authors · 2024-11-19
> **Response to Reviewer avyx (2/2)**
>
> >Weakness: Analyzing why certain datasets/attack combinations benefit more from the approach. By incorporating such an angle, the contribution could be significantly broadened.
>
> Thanks for your valuable suggestion. In the revised paper, we supplement discussion about this topic in experiments (Section 4.2). We show the performance boosts achieved by OMS are significantly influenced by the strength of the original attacker. Specifically, **when the original attacker is relatively weak, the additional improvements provided by OMS are more pronounced.** Our method can significantly augment the performance of weak attackers (Table 1, 20.77 in AUC for SecMI on LFW). This may be attributed to the fact that when the original attacker struggles to identify the samples, the additional information provided by OMS becomes more significant.
>
> >Weakness: Presentation Issues
>
> We apologize for the presentation issues in our previous submission. We have taken steps to improve the presentation of our work. Specifically, we have revised Table 2 to include the percentage improvement in performance, which provides a clearer comparison and facilitates a more informed understanding of the effectiveness of our approach. We have corrected the citation commands from “\cite” to “\citep” and “\citet” and revised the abbreviation usage throughout the paper to improve clarity and readability. In addition, we have added a brief introduction for the fixed-point iteration process. To further support this introduction, we have provided additional preliminaries on fixed-point iteration in Appendix A. We believe that these improvements enhance the presentation of our work.
>
> >Weakness: What does it mean: "During the inference phase, due to the infeasibility of the training noise" $\rightarrow$ what is infeasible? Getting access to the original noise?
>
> Yes. In the context of our work, we aim to emphasize the challenge of accessing the original noise used during the training stage.

---

> > ### Comment · Reviewer_avyx · 2024-11-24
> > **Thank you to the authors for their response**
> >
> > Thank you to the authors for addressing the comments.
> >
> >
> > The insights added to F1 are interesting, as they indeed visually show an improvement of separability. The reported numbers and percentages are not entirely clear to me, though. How is the threshold chosen that leads to the conclusion that 1795 nonmembers are now correctly classified as members?
> >
> > I also appreciated the analysis in Section 4.2. Yet, it raises the further question: why would one have interest in improving weak attacks? SecMI + the addition of the work OMS reach 86.48 where PIA alone without OMS reaches 88.75 on CIFAR10. Still, the strong attacks also face improvements, which are better highlighted with the new formatting.
> >
> > I acknowledge the answer on the raised limitation of the method, namely that it claims to identify is the convergence speed, rather than the loss itself that helps in distinguishing members from nonmembers. This implicitly underlies already the original MIAs which just use the first noise step, rather than doing the first+second as done in this work. I, therefore, remain with my score.

---

> > > ### Author Response · Authors · 2024-11-25
> > > **Response to the further feedback of Reviewer avyx (1/2)**
> > >
> > > Thank you for reading our responses and providing further feedback! We have uploaded the revised paper (highlighted in red) incorporated further analysis on impacts OMS concerning different samples (Appendix F, Line 993-995) and different original attacks (Section 4.2, Line 369-371) according to your advice, which we believe will significantly improve the contribution of our work. Here, we would like to further clarify our contribution as follows:
> > >
> > > **1. Introduction of the novel noise searching framework.**
> > >
> > > Our work introduces the concept of **“noise step”**, and no previous work has explicitly defined or utilized the “noise step” concept. More concretely, **“noise step” is a concept rooted in our proposed fixed-point iteration framework**, which is a novel perspective for diffusion MIA (acknowledged by Reviewer rNW3 “creative reformulation”, “fresh perspective” and Reviewer tTKh “unique approach” “novel formulation” “fresh idea”) within the proposed novel fixed-point iteration framework. Previous loss-based methods [1-3] rely on reconstruction loss, assuming that diffusion models exhibit better reconstruction for members. We point out the **noise inconsistency issue** and demonstrate that the loss is not the key factor, which you also agree with. Thus, **while loss-based methods may be effective, their underlying assumptions are flawed.** In contrast, **we explain the effectiveness by convergence dynamics,** which has swept away the dark clouds floating over loss-based methods, which **we believe is more important than proposing a specific MIA loss metric for diffusion models.**
> > >
> > > **2. Thorough analysis of convergence properties.**
> > >
> > > We conduct a comprehensive analysis of the convergence dynamics of members and nonmembers within our proposed fixed-point iteration framework. This includes validating the **convergence properties** using **Jacobian matrix and the contraction mapping theorem** [4], which provides a solid theoretical foundation for our method. Furthermore, we analyze **the convergence speed** through **residual and error analysis**. Your acknowledgement of differences in convergence dynamics between members and nonmembers underscores the significance of our contributions. **These analyses represent important additions to the understanding of diffusion model MIAs and should not be overlooked.**
> > >
> > > **3. Significant performance improvements with our proposed OMS.**
> > >
> > > While concerns may arise regarding the potential influence of existing MIA methods, experimental results demonstrate that our proposed OMS, **not marginally, but significantly boosts the performance of various MIA methods** (loss method [1-3], quantile method [5,6], gradient method [7]) across different datasets. We have also conducted experiments on a range of diffusion models, including CNN-based and Transformer-based architectures. Notably, we are the **first** to demonstrate the practical effectiveness of MIAs on **Transformer-based diffusion models**. This contribution is significant as it highlights the potential and practicality of both existing MIAs and our proposed OMS in transformer-based architectures.
> > >
> > > We sincerely hope you can reconsider your rating score and we are open to answering any further questions you may have.

---

> > > > ### Comment · Reviewer_avyx · 2024-12-01
> > > > **Question**
> > > >
> > > > > Notably, we are the first to demonstrate the practical effectiveness of MIAs on Transformer-based diffusion models. This contribution is significant as it highlights the potential and practicality of both existing MIAs and our proposed OMS in transformer-based architectures.
> > > >
> > > > What does "practical effectiveness" refer to?
> > > >
> > > > U-ViT seems to be trained on the CIFAR10 (Table 5), which is rather toy dataset in terms of size and complexity. Therefore, I am not sure we can be discussing practical effectiveness based on the results.
> > > >
> > > >
> > > > Why would we expect transformer-based models to behave significantly different from non-transformer ones?

---

> > > > > ### Author Response · Authors · 2024-12-02
> > > > > **Response_v2 to the further feedback of Reviewer avyx (1/2)**
> > > > >
> > > > > > Why would we expect transformer-based models to behave significantly different from non-transformer ones?
> > > > >
> > > > > Thank you for your question regarding the expected differences in behavior between transformer-based models and non-transformer models. We appreciate the opportunity to clarify our rationale.
> > > > >
> > > > > Transformer-based models, such as Vision Transformers (ViTs), fundamentally differ from non-transformer models, like Convolutional Neural Networks (CNNs), in their architectural design and processing mechanisms. ViTs are centered around attention mechanisms and Feed-Forward Networks (FFNs), which enable them to process image patches through attention layers. This structural distinction is crucial for understanding their behavior in membership inference attacks (MIAs).
> > > > >
> > > > > Firstly, the patchification operation inherent in ViTs, which is skilled at handling high-level tasks (e.g. classification), may influence the prediction of noise in diffusion models, a process that is more aligned with low-level image generation tasks [1]. This discrepancy suggests that ViTs might handle noise and image reconstruction differently than CNNs, which could impact their susceptibility to MIAs.
> > > > >
> > > > > Secondly, the attention mechanism in ViTs allows for the capture of global spatial information across the entire image, which is a capability that surpasses the local receptive fields of CNNs. This global awareness empowers ViTs a lower inductive bias, making them more adept at producing generalized models [2]. Consequently, ViTs are expected to be more robust against MIAs due to their ability to generalize better.
> > > > >
> > > > > Current research [3] indicates that ViT **classifiers** exhibit greater robustness against MIAs compared to CNN **classifiers**, as evidenced by overlapping Top-1 Confidence scores for both members and nonmembers. The lack of significant magnitude difference in adversarial perturbations between members and nonmembers implies that the margin to the decision boundary is similar for both, rendering adversarial-based MIAs [4] less effective for ViT classifiers.
> > > > >
> > > > > Furthermore, to our knowledge, no prior MIA has demonstrated effectiveness on ViT-based **generative models** such as GANs, VAEs, or diffusion models. Given the distinct modeling approaches between ViTs and CNNs, and the unsatisfactory performance of MIAs on ViT classifiers, there is a clear research gap: the effectiveness of current Diffusion MIAs, which rely on reconstruction loss assumptions, has not been established for ViT-based models. This gap is significant as it leaves open questions about the potential and effectiveness of current methods and their applicability to ViT-based generative models.
> > > > >
> > > > > To address this gap, we have conducted experiments on U-ViT, implying that ViT architecture is more robust than CNN in diffusion models (as shown in Tables 1 and 4 in the main text). For your convenience, we also provide a direct comparison of DDPM and U-ViT in the following tables.
> > > > >
> > > > > Our observations indicate that current MIA methods [5-7] exhibit reduced efficacy in the context of U-ViT. Specifically, PIA [7] demonstrates an AUC of 52.91, which approximates the level of random guessing. This phenomenon is attributed to the noise deviation from the standard Gaussian distribution produced by PIA. To address this issue, we employ a normalization technique to align the noise with the standard Gaussian distribution (denoted as PIAN). Additionally, our proposed OMS strategy is observed to enhance the performance of current MIA methods in both CNNs and ViTs, further validating the efficacy of our proposed method.
> > > > >
> > > > > We will incorporate these discussions in the revised paper to emphasize the behavior difference across different architectures. We hope this response adequately addresses your question.

---

> > > > > > ### Comment · Reviewer_avyx · 2024-12-02
> > > > > > **Thank you**
> > > > > >
> > > > > > > We will incorporate these discussions in the revised paper to emphasize the behavior difference across different architectures. We hope this response adequately addresses your question.
> > > > > >
> > > > > > This seems indeed like a valuable discussion and the paper will benefit from adding it.

---

> > > > > ### Author Response · Authors · 2024-12-02
> > > > > **Response_v2 to the further feedback of Reviewer avyx (2/2)**
> > > > >
> > > > > > What does "practical effectiveness" refer to? U-ViT seems to be trained on the CIFAR10 (Table 5), which is rather toy dataset in terms of size and complexity. Therefore, I am not sure we can be discussing practical effectiveness based on the results
> > > > >
> > > > > Thanks for your comment. We understand your skepticism about the dataset's size and complexity and appreciate the opportunity to provide further clarification. The term "practical" in our context refers to the ability of our model to perform well under controlled conditions that simulate real-world scenarios, as well as its potential to be generalized to more complex datasets.
> > > > >
> > > > > Besides, we would like to clarify our rationale for using Cifar10. First, Cifar10, despite its size, is a well-established benchmark in the field of computer vision. Its structure and diversity allow us to develop and rigorously test new algorithms. Second, the extensive use of CIFAR-10 in the literature allows us to compare our findings with a broad range of existing studies, ensuring that our results are not only reproducible but also contribute to the ongoing discourse in the field.
> > > > >
> > > > > Thank you for your helpful comments again. We are currently conducting experiments on more complex datasets (ImageNet). We will ensure that these results are made available as soon as they are ready, to address your concerns and to provide a more complete picture of our model's performance.
> > > > >
> > > > > ## Reference
> > > > > [1] Bao, Fan, et al. "All are worth words: A vit backbone for diffusion models." CVPR 2023.
> > > > >
> > > > > [2] Oh, Seungeun, et al. "Privacy-Preserving Split Learning with Vision Transformers using Patch-Wise Random and Noisy CutMix." TMLR 2024.
> > > > >
> > > > > [3] Boyu Zhang, et al. “Membership Inference Attacks against Vision Transformers: Mosaic MixUp Training to the Defense” CCS 2024.
> > > > >
> > > > > [4] Choquette-Choo, Christopher A., et al. "Label-only membership inference attacks." International conference on machine learning. ICML, 2021.
> > > > >
> > > > > [5] Matsumoto et al. “Membership inference attacks against diffusion models.” Arxiv 2023.
> > > > >
> > > > > [6] Duan, Jinhao, et al. "Are diffusion models vulnerable to membership inference attacks?." ICML 2023.
> > > > >
> > > > > [7] Kong, Fei, et al. "An Efficient Membership Inference Attack for the Diffusion Model by Proximal Initialization." ICLR 2024.

---

> > > > > > ### Comment · Reviewer_avyx · 2024-12-02
> > > > > > **A valuable addition**
> > > > > >
> > > > > > > Thank you for your helpful comments again. We are currently conducting experiments on more complex datasets (ImageNet). We will ensure that these results are made available as soon as they are ready, to address your concerns and to provide a more complete picture of our model's performance.
> > > > > >
> > > > > > I agree that this will be a valuable addition to the paper. When integrating more complex datasets, it will also be important to avoid the pitfalls of evaluation, as, for example, mentioned in [1].
> > > > > >
> > > > > >
> > > > > >
> > > > > > [1] Dubiński, Jan, Antoni Kowalczuk, Stanisław Pawlak, Przemyslaw Rokita, Tomasz Trzciński, and Paweł Morawiecki. "Towards more realistic membership inference attacks on large diffusion models." In Proceedings of the IEEE/CVF Winter Conference on Applications of Computer Vision, pp. 4860-4869. 2024.

---

> ### Author Response · Authors · 2024-11-25
> **Response to the further feedback of Reviewer avyx (2/2)**
>
> > How is the threshold chosen that leads to the conclusion that 1795 nonmembers are now correctly classified as members?
>
> We choose the threshold based on the implementation of previous studies (i.e., SecMI and PIA [8-9]). A detailed explanation of the threshold selection process is provided in the original submission (Appendix E). Specifically, “we randomly select a small fraction (10%) of samples to obtain an optimal threshold, which is then used to classify the remaining samples”. For the reported percentage, we test 50,000 samples on Cifar10 dataset, consisting of 25,000 member and 25,000 nonmember samples. This dataset and the experimental setup are described in detail in Appendix C of the original submission.
>
> > Why would one have interest in improving weak attacks?
>
> We want to clarify that the term **‘weak’** in this context refers to **inferior performance**, rather than a fundamental weakness in the method itself. **Attackers with strong performance exhibit limitations in other perspective.** For instance, GSA [7] requires access to the model’s gradients, positioning it as a purely white-box method with strong performance. The behavior of PIA [2] to predict proximal points is abnormal and distinct from typical user behavior. SecMI [1] requires over ten queries to the target model, whereas NA [3] requires only one, hence SecMI's stronger performance.
>
> In practical applications, **the choice of attacker should be tailored to the specific scenario, taking into account the unique strengths and weaknesses of each MIA method.** While performance is an important consideration, it should not be the sole criterion for selecting an attacker. Furthermore, as noted, our method can also enhance strong attackers. Therefore, we believe that it is not only important to further improve the performance of MIA with high accuracy but also to enhance weak attackers, as this holds practical significance in various contexts.
>
> ## Reference
>
> [1] Duan, Jinhao, et al. "Are diffusion models vulnerable to membership inference attacks?." ICML 2023.
>
> [2] Kong, Fei, et al. "An Efficient Membership Inference Attack for the Diffusion Model by Proximal Initialization." ICLR 2024.
>
> [3] Matsumoto et al. “Membership inference attacks against diffusion models.” Arxiv 2023.
>
> [4] Berinde and Takens. “Iterative approximation of fixed points.” Springer 2007.
>
> [5] Bertran et al. “Scalable membership inference attacks via quantile regression.” NeurIPS 2024.
>
> [6] Tang et al. “Membership inference attacks on diffusion models via quantile regression.” Arxiv 2023.
>
> [7] Pang et al. “White-box membership inference attacks against diffusion models.”
>
> [8] https://github.com/jinhaoduan/SecMI
>
> [9] https://github.com/kong13661/PIA

---

> ### Author Response · Authors · 2024-12-01
> **Discussion period is ending soon**
>
> Dear Reviewer,
>
> We hope to have resolved all your concerns. If you have any further comments, we will be happy to address them before the rebuttal period ends. If there are none, then we would appreciate it if you could reconsider your rating.
>
> Regards,
>
> Authors

---

> ### Author Response · Authors · 2024-12-02
> **Tables**
>
> Comparison of DDPM-Cifar10 and U-ViT-Cifar10
> |               | #Params | #Epochs |
> |:-------------:|:-------:|:-------:|
> |  DDPM-Cifar10 |   36M   |   1000  |
> | U-ViT-Cifar10 |   44M   |   1280  |
>
>        Results on DDPM-Cifar10:
> |       |               |  DDPM-Cifar10  |                |
> |:-----:|:-------------:|:--------------:|:--------------:|
> |       |      ASR      |       AUC      |    TPR@1%FPR   |
> |   NA  |     71.86     |      78.28     |      6.42      |
> |  +OMS |     78.21     |      85.71     |      12.12     |
> | $\Delta \uparrow$ | +6.35(+8.84%) |  +7.43(+9.49%) |  +5.70(+88.8%) |
> | SecMI |     84.01     |      90.68     |      9.15      |
> |  +OMS |     86.48     |      92.82     |      15.87     |
> | $\Delta \uparrow$ | +2.47(+2.94%) |  +2.14(+2.36%) |  +6.72(+73.4%) |
> |  PIA  |     88.75     |      94.89     |      28.86     |
> |  +OMS |     91.78     |      97.26     |      60.11     |
> | $\Delta \uparrow$ | +3.03(+3.41%) |  +2.37(+2.50%) |  31.25(+108%)  |
>
>       Results on U-ViT-Cifar10:
> |       |               |  U-ViT-Cifar10 |                |
> |:-----:|:-------------:|:--------------:|:--------------:|
> |       |      ASR      |       AUC      |    TPR@1%FPR   |
> |   NA  |     61.47     |      63.66     |      2.62      |
> |  +OMS |     68.51     |      74.31     |      8.65      |
> | $\Delta \uparrow$ | +7.04(+11.5%) | +10.65(+16.7%) |  +6.03(+230%)  |
> | SecMI |     68.21     |      74.44     |      12.88     |
> |  +OMS |     74.95     |      82.31     |      24.35     |
> | $\Delta \uparrow$ | +6.74(+9.88%) |  +7.87(+10.6%) | +11.47(+89.1%) |
> |  PIAN |     59.36     |      61.13     |      3.42      |
> |  +OMS |     69.11     |      75.42     |      9.86      |
> | $\Delta \uparrow$ | +9.75(+16.4%) | +14.29(+23.4%) |  +6.44(+188%)  |
> |  PIA  |     54.60     |      52.91     |      1.80      |

---

> ### Author Response · Authors · 2024-12-02
> **The experimental results are available!**
>
> To substantiate the efficacy of our approach on transformer-based diffusion models, we have extended our experimental analysis to encompass the U-ViT model, which was trained on the ImageNet dataset. We utilize the pre-trained U-ViT checkpoint (U-ViT-M/4) sourced from [1] directly. Specifically, we employ 10,000 samples from the ImageNet training set as members and 10,000 samples from the ImageNet validation set as nonmembers. The outcomes of these experiments are presented in the subsequent table.
> |         |      AUC      |      ASR      |   TPR@1%FPR   |
> |:-------:|:-------------:|:-------------:|:-------------:|
> |    NA   |     65.84     |     67.85     |      4.00     |
> |  NA+OMS |     68.70     |     70.25     |      6.20     |
> |    $\Delta \uparrow$    | +2.86(+4.34%) | +2.40(+3.54%) |  +2.20(+55%)  |
> |   PIA   |     63.35     |     61.14     |      4.10     |
> | PIA+OMS |     65.25     |     65.23     |      5.95     |
> |    $\Delta \uparrow$    | +1.90(+3.00%) | +4.09(+6.69%) | +1.85(+31.4%) |
>
> Our findings indicate that our method can significantly enhance current MIAs on transformer-based diffusion models when applied to more complex datasets, thereby further validating the effectiveness of our approach.
>
> [1] https://github.com/baofff/U-ViT
>
> -----------------------------------------------------
> Thank you for your helpful comments again. We hope to have resolved all your concerns. If you have any further comments, we will be happy to address them before the rebuttal period ends.

---

> ### Author Response · Authors · 2024-12-02
> **Thanks for your advice!**
>
> We thank the reviewer for highlighting the importance of avoiding evaluation pitfalls, as discussed in [1]. We acknowledge the significance of this concern and are testing the performance of our method towards Stable Diffusion on the LAION-mi dataset to ensure rigorous evaluation. Given the impending end of the discussion period and the computational overload required for LAION-mi dataset (the deduplication and sanitization preprocess), we may not be able to provide these results before the deadline. However, we will endeavor to do so as soon as possible. If we are unable to meet this timeline, we will include the performance evaluation leveraging the LAION-mi dataset in the revised paper.
>
> (Besides, the experimental results on U-ViT-ImageNet are available.)
>
> Thank you again for your valuable feedback. If you have any further comments, we will be happy to address them before the rebuttal period ends.
>
> [1] Dubiński, Jan, Antoni Kowalczuk, Stanisław Pawlak, Przemyslaw Rokita, Tomasz Trzciński, and Paweł Morawiecki. "Towards more realistic membership inference attacks on large diffusion models." In Proceedings of the IEEE/CVF Winter Conference on Applications of Computer Vision, pp. 4860-4869. 2024.

---

> > ### Author Response · Authors · 2024-12-04
> > **The experimental results on LAION-mi datasets are available!**
> >
> > We conducted additional experiments on the LAION-mi datasets to further validate the efficacy of our method. Due to time limitations, for the non-member samples of LAION-mi, we obtain 500 images in total by deduplicating and sanitizating the LAION-2B Multi Translated, following the preprocessing procedure outlined in [1]. The results are presented in the table below:
> >
> > |       |               | SD1.5-LAION-mi |              |
> > |:-----:|:-------------:|:--------------:|:------------:|
> > |       |      AUC      |       ASR      |   TPR@1%FPR  |
> > |   NA  |     59.98     |      62.10     |     4.20     |
> > |  +OMS |     62.95     |      65.30     |     4.70     |
> > | $\Delta \uparrow$ | +2.97(+4.95%) |  +3.20(+5.15%) | 0.50(+11.9%) |
> >
> > These results demonstrate that our method can enhance existing MIA methods across various experimental settings, including those with and without distribution shifts. We hope this addresses your concerns.
> >
> > [1] Dubiński, Jan, Antoni Kowalczuk, Stanisław Pawlak, Przemyslaw Rokita, Tomasz Trzciński, and Paweł Morawiecki. "Towards more realistic membership inference attacks on large diffusion models." In Proceedings of the IEEE/CVF Winter Conference on Applications of Computer Vision, pp. 4860-4869. 2024.

---

### Official Review · Reviewer_tTKh · 2024-11-02

**Soundness:** 3
**Presentation:** 3
**Contribution:** 3
**Rating:** 6
**Confidence:** 4

**Summary:**

This paper introduces One More Step (OMS) Noise Searching, a framework to enhance Membership Inference Attacks (MIA) on diffusion models. Traditional MIA approaches rely on prediction loss differences between members and non-members, which can be ineffective for diffusion models due to random noise sampling during training. The authors propose reformulating MIA as a noise search problem, aiming to identify training noise that corresponds to specific data records. By using fixed-point iteration, the proposed method iteratively searches for this noise, leveraging the observation that member data tends to converge faster than non-member data. This process also introduces OMS, a refinement that incorporates an additional iteration to improve discrimination between member and non-member records. Experimental results on various diffusion models show that OMS significantly boosts MIA performance.

**Strengths:**

1.The paper proposes a unique approach to handling noise in diffusion models by shifting the MIA focus from prediction loss to noise searching. This formulation is novel within MIA related research and introduces fresh ideas on addressing privacy risks specific to diffusion models.

2.The study is thorough, with extensive experiments across different types of diffusion models (e.g., CNN-based, Transformer-based) and datasets. This breadth strengthens the paper’s empirical validity.

3.The paper is well-organized, with detailed explanations of the problem, methodology, and fixed-point iteration approach.

**Weaknesses:**

1.While the fixed-point iteration method is interesting, the paper could benefit from a deeper exploration of its convergence properties

2.Additional analysis on scalability and time complexity could make the contribution more robust.

**Questions:**

1.I’m curious about the computational requirements of the OMS approach. Could the authors provide more details on how OMS performs in terms of memory and computation time, especially when additional steps are taken?

2.Could the authors clarify how the convergence rate differs between member and non-member samples, and whether this could be generalized across various types of diffusion models? More insights into this mechanism would help to strengthen the theoretical foundation of the approach.

I am giving a score of 6 (marginally above the acceptance threshold), as the novelty and contribution appear promising and potentially meet the standards for ICLR. I will consider the expertise and feedback of other reviewers, particularly those with a stronger background in diffusion models, and may adjust my score based on additional insights or clarifications.

---

> ### Author Response · Authors · 2024-11-19
> **Response to Reviewer tTKh**
>
> Thank you for taking the time to review our paper, and for your helpful comments. Below we will address your concerns point by point:
>
> >Weakness: While the fixed-point iteration method is interesting, the paper could benefit from a deeper exploration of its convergence properties.
>
> Thanks for this suggestion! We have enhanced our exploration of the convergence properties in the revised paper. In the original submission, we discussed the convergence of the generated sequence $\\{\\epsilon^n\\}$ and leveraged residuals as an indicator of convergence speed. To further deepen our analysis, **we have now included an examination of the function $f$** which is composed by the diffusion models. Specifically, **we have analyzed the contraction of the objective function** and estimated the contraction constant $k$, observing that it remains continuously below 1, thereby reinforcing the convergence of the fixed-point iteration (More details can be found in Appendix B.1). Additionally, **we have conducted an error analysis according to [1] to investigate the convergence speed**, providing compelling evidence that members converge more rapidly than non-members (More details can be found in Appendix B.2).
>
> >Weakness & Question: More details on how OMS performs in terms of memory and computation time.
>
> Thanks for this comment. In terms of memory and computation time, our proposed **OMS method exhibits comparable performance to baseline attackers.** Specifically, our method requires one additional query to the target diffusion model compared to the baseline methods. Consequently, in terms of memory consumption, **our method utilizes the same amount of memory as the baseline attackers.** Regarding computation efficacy, **the increase in the number of queries translates to a minor increase in computation time.** We provide the required queries for baseline attackers:
>
> | Queries  | NA  | SecMI | PIA |
> |----------|-----|-------|-----|
> | w/o OMS  | 1   | 12    | 2   |
> | with OMS | 1+1 | 12+1  | 2+1 |
>
> >Question: Could the authors clarify how the convergence rate differs between member and non-member samples, and whether this could be generalized across various types of diffusion models? More insights into this mechanism would help to strengthen the theoretical foundation of the approach.
>
> Thanks for the comments. We have conducted a more detailed analysis of the convergence rate differences between member and nonmember samples, and their generalization across various types of diffusion models. Specifically, **we have included an error analysis in Appendix B.2**, which provides an additional perspective on the convergence speed. Furthermore, **we have conducted a convergence analysis of the transformer-based diffusion model (U-ViT)**, using both residual and error analysis, and presented the results in Figure B.2. These findings provide strong evidence that the assumption that **member samples converge faster can be generalized across various types of diffusion models**. This analysis helps to strengthen the theoretical foundation of our approach and provides deeper insights into the underlying mechanisms of convergence in fixed-point iteration.
>
> [1] Berinde, V. "Iterative approximation of fixed points." (2007).

---

> > ### Comment · Area_Chair_PEij · 2024-11-26
> >
> > Dear reviewer tTKh,
> >
> > Could you please response to authors' rebuttal and see if you would like to update your review? Thanks very much!
> >
> > AC

---

> ### Author Response · Authors · 2024-12-01
> **Discussion period is ending soon**
>
> Dear Reviewer,
>
> We hope to have resolved all your concerns. If you have any further comments, we will be happy to address them before the rebuttal period ends. If there are none, then we would appreciate it if you could reconsider your rating.
>
> Regards,
>
> Authors

---

### Official Review · Reviewer_rNW3 · 2024-11-03

**Soundness:** 3
**Presentation:** 3
**Contribution:** 3
**Rating:** 6
**Confidence:** 4

**Summary:**

In this paper, the authors explore privacy risks in diffusion models by improving MIA. Traditional MIA methods are ineffective for diffusion models due to noise in training. This paper proposes a novel noise search approach called OMS (One More Step), which refines MIA by adding a fixed-point iteration step to better distinguish between training members and non-members. This method significantly enhances MIA accuracy across various diffusion models and datasets, addressing privacy concerns in data-intensive AI models.

**Strengths:**

+. The paper introduces a creative reformulation of Membership Inference Attacks (MIA) as a noise search problem, which is a fresh perspective on tackling privacy risks associated with diffusion models.
+. The proposed One More Step (OMS) enhancement to existing MIA techniques effectively improves the accuracy of identifying members in diffusion models. This additional fixed-point iteration step is a simple yet impactful modification that leverages the specific characteristics of diffusion processes.
+. The authors conducted experiments across various diffusion models and datasets, providing a robust demonstration of OMS’s effectiveness.

**Weaknesses:**

-. Some statements in the paper can lead to confusion. For example, in the introduction, it is stated: “To address these privacy concerns, Membership Inference Attacks (MIA) Shokri et al. (2017) have emerged as a potential solution.” This phrasing is misleading, as MIAs themselves are privacy attacks, not solutions to privacy concerns. This could be clarified to avoid misunderstanding.
-. While the methodology of this paper is technically sound, it would benefit from a clearer explanation of the real-world implications of privacy risks in diffusion models. To strengthen the paper, it is recommended that the authors include a specific section discussing the real-world impacts of privacy risks and their potential effects on individuals or organizations, along with 1-2 concrete examples to support this discussion.
-. Some terms and technical concepts, such as “fixed-point iteration” and “convergence rate,” are introduced without sufficient background or explanation. For readers unfamiliar with these mathematical concepts, a brief definition or background could make the paper more accessible.

**Questions:**

Q1: Could the authors elaborate on how the noise search mechanism operates in practice? Specifically, what criteria are used to determine the optimal stopping point in the noise search process?
Q2: How well does OMS perform across different types of diffusion models, beyond those tested in the experiments? Are there specific classes of diffusion models where OMS may be less effective?

---

> ### Author Response · Authors · 2024-11-19
> **Response to Reviewer rNW3 (1/2)**
>
> Thank you for taking the time to review our paper, and for your helpful comments. Below we will address your concerns point by point:
>
> >Weakness: Some statements in the paper can lead to confusion. For example, in the introduction, it is stated: “To address these privacy concerns, Membership Inference Attacks (MIA) Shokri et al. (2017) have emerged as a potential solution.”
>
> Thank you for pointing this out. In response to the potential confusion arising from the statement in the introduction, we have revised the corresponding part to enhance clarity and accuracy. Specifically, we have replaced “To address these privacy concerns” with “To audit these privacy risks”.
>
> >Weakness: While the methodology of this paper is technically sound, it would benefit from a clearer explanation of the real-world implications of privacy risks in diffusion models. To strengthen the paper, it is recommended that the authors include a specific section discussing the real-world impacts of privacy risks and their potential effects on individuals or organizations, along with 1-2 concrete examples to support this discussion.
>
> Thanks for this advice! To give a clearer explanation of the privacy risks in diffusion models, **we have expanded our discussion in Appendix H** to provide a comprehensive analysis of the practical applications of MIA and the associated privacy risks. Specifically, we elaborate how MIA can serve as a powerful tool to detect unauthorized data abuse, which is crucial in the current digital landscape. This also raises awareness of data privacy concerns. Furthermore, we highlight the potential utility of MIA for organizations, as it can be leveraged as a privacy measurement and employed for auditing and regulatory purposes.
>
> >Weakness: Some terms and technical concepts are introduced without sufficient background or explanation.
>
> Thanks for this advice. To enhance the clarity and accessibility of our paper, we have **included a brief description of the fixed-point iteration process in the introduction (lines 70-72)** to provide readers with a foundational understanding of this key concept. Additionally, **we have appended a new section (Appendix A)**, introducing the preliminaries of the fixed-point iteration, which offers a more comprehensive overview of the technical details.
>
> >Question: Could the authors elaborate on how the noise search mechanism operates in practice? Specifically, what criteria are used to determine the optimal stopping point in the noise search process?
>
> When to stop is a trade-off between effectiveness and computational costs. We have conducted experiments to analyze the impact of iteration times on the performance of the noise search mechanism. As demonstrated in Figure 3 of the main text, two iterations yield significant performance gains compared to one iteration (85.71 vs 78.28). However, the gains observed with three iterations compared to two iterations are relatively limited (87.06 vs 85.71). This experimental evidence suggests that **while additional iterations may further enhance performance, stopping after two iterations represents a computationally economical and efficient choice.** Furthermore, we provide additional ablations on iteration numbers (N) on Cifar10 dataset. The results are in the following table which show that further steps beyond the proposed OMS (N=2) do not result in significant performance gains, indicating a rapid performance saturation.
> |     | N=1   | N=2   | N=3   | N=4   | N=5   |
> |-----|-------|-------|-------|-------|-------|
> | NA  | 78.28 | 85.71 | 87.06 | 87.84 | 87.93 |
> | PIA | 94.89 | 97.26 | 98.04 | 98.42 | 98.35 |
>
> Therefore, based on these findings, stopping when N=2 is a practically viable choice for determining the optimal stopping point in the noise search process.

---

> ### Author Response · Authors · 2024-11-19
> **Response to Reviewer rNW3 (2/2)**
>
> >Question: How well does OMS perform across different types of diffusion models, beyond those tested in the experiments?
>
> Our experiments in the original submission demonstrate that **the proposed method exhibits strong generalization capabilities across different architectures** (including conditional and unconditional, CNN-based and transformer-based diffusion models).  Additionally, **we have conducted further experiments on fine-tuned diffusion models** to assess its performance in more specific contexts. Specifically, we evaluate our method on diffusion models finetuned on the Pokemon dataset following previous works [1-2]. The results of this experiment, as presented below, further validate the effectiveness of the OMS method.
>
> | AUC      | NA    | SecMI | PIA   |
> |----------|-------|-------|-------|
> | w/o OMS  | 77.71 | 61.25 | 64.67 |
> | with OMS | 79.28 | 65.98 | 67.04 |
> | $\Delta \uparrow$        | 1.57  | 4.73  | 2.37  |
>
> >Question: Are there specific classes of diffusion models where OMS may be less effective?
>
> We have not identified any specific classes of diffusion models where the OMS may be less effective. However, we have observed that the performance gains achieved by OMS are influenced by the strength of the original attacker. Specifically, **when the original attacker is relatively weak, the additional improvements provided by OMS are more pronounced**. Our method can significantly augment the performance of weak attackers (Table 1, 20.77 in AUC for SecMI on LFW). Even for nearly saturated attackers, our method can achieve considerable improvements (Table 1, 94.89 to 97.26 in AUC for PIA on Cifar10).   In summary, OMS demonstrates robust performance across a wide range of diffusion models, and is particularly effective for weak attackers.
>
> [1] Duan, Jinhao, et al. "Are diffusion models vulnerable to membership inference attacks?." ICML 2023.
>
> [2] Kong, Fei, et al. "An Efficient Membership Inference Attack for the Diffusion Model by Proximal Initialization." ICLR 2024.

---

> > ### Comment · Reviewer_rNW3 · 2024-11-25
> >
> > Thank you very much for the author's comprehensive response, which has partially alleviated my concerns. Therefore, I will maintain my current rating.

---

> > > ### Author Response · Authors · 2024-11-25
> > > **Thank you for reading our responses!**
> > >
> > > Thank you for your continued support of our work. We're glad that our response has alleviated some of your concerns. We're also grateful for the chance to delve into the issues you still have. Your insights are invaluable to us, and we would be very appreciative if you could share your remaining concerns. Could you kindly inform us of any remaining concerns you have?

---

> ### Author Response · Authors · 2024-12-01
> **Discussion period is ending soon**
>
> Dear Reviewer,
>
> We hope to have resolved all your concerns. If you have any further comments, we will be happy to address them before the rebuttal period ends. If there are none, then we would appreciate it if you could reconsider your rating.
>
> Regards,
>
> Authors

---

### Author Response · Authors · 2024-11-19
**General Response by Authors**

We express our gratitude to all the reviewers for their insightful and constructive feedback. Each reviewer's comments have been individually addressed and responded to.

Additionally, we have uploaded the revised manuscript, incorporating the following key revisions:

+ We have conducted a more thorough analysis of the convergence property of the fixed-point iteration process using the contraction mapping theorem (Appendix B.1).

+ We have further examined the difference in convergence speed between members and nonmembers through error analysis (Appendix B.2).

+ We have validated the convergence property and convergence speed in transformer-based diffusion models (Appendix B.3).

+ We provide an intuition of fixed-point iteration in introduction, and more preliminaries on the subject in Appendix A.

+ We have explored whether there are samples that benefits disproportionately from the proposed OMS (see Appendix F).

+ We have discussed the implications of MIA for diffusion models (Appendix H).

Lastly, we would appreciate all reviewers' time again. Would you mind checking our response and confirming whether you have any further questions? **We are anticipating your post-rebuttal feedback!**

---

### Meta-Review · Area_Chair_PEij · 2024-12-23

**Metareview:**

The paper is borderline. Reviewers generally agreed that there are some interesting contributions in the paper, but it needs more improvements before publication.

Strength: the proposed method appeared to somewhat novel.
Weakness:
1. Presentation of the paper needs some improvement at least two reviewers found the paper to be confusing in several aspects.
2. Additional experiments and insights could be useful to strengthen the paper.

**Additional Comments On Reviewer Discussion:**

The overall support from the reviewers is lukewarm and not strong. During the discussions the authors addressed most of concerns but didn't convince majority of the reviewers to change their recommendations. During the post-rebuttal discussion, the reviewers came to the conclusion that the paper was not ready for publication due to the competitiveness of the ICLR.

---

### Decision · Program_Chairs · 2025-01-22

Reject